# Learning Proximal Operators to Discover Multiple Optima

**Lingxiao Li**
MIT CSAIL
lingxiao@mit.edu

**Noam Aigerman**
Adobe Research
aigerman@adobe.com

**Vladimir G. Kim**
Adobe Research
vokim@adobe.com

**Jiajin Li**
Stanford University
jiajinli@stanford.edu

**Kristjan Greenewald**
IBM Research, MIT-IBM Watson AI Lab
kristjan.h.greenewald@ibm.com

**Mikhail Yurochkin**
IBM Research, MIT-IBM Watson AI Lab
mikhail.yurochkin@ibm.com

**Justin Solomon**
MIT CSAIL
jsolomon@mit.edu

## Abstract

Finding multiple solutions of non-convex optimization problems is a ubiquitous yet challenging task. Most past algorithms either apply single-solution optimization methods from multiple random initial guesses or search in the vicinity of found solutions using ad hoc heuristics. We present an end-to-end method to learn the *proximal operator* of a family of training problems so that multiple local minima can be quickly obtained from initial guesses by iterating the learned operator, emulating the *proximal-point algorithm* that has fast convergence. The learned proximal operator can be further generalized to recover multiple optima for unseen problems at test time, enabling applications such as object detection. The key ingredient in our formulation is a proximal regularization term, which elevates the convexity of our training loss: by applying recent theoretical results, we show that for weakly-convex objectives with Lipschitz gradients, training of the proximal operator converges globally with a practical degree of over-parameterization. We further present an exhaustive benchmark for multi-solution optimization to demonstrate the effectiveness of our method.

## 1 Introduction

Searching for multiple optima of an optimization problem is a ubiquitous yet under-explored task. In applications like low-rank recovery (Ge et al., 2017), topology optimization (Papadopoulos et al., 2021), object detection (Lin et al., 2014), and symmetry detection (Shi et al., 2020), it is desirable to recover multiple near-optimal solutions, either because there are many equally-performant global optima or due to the fact that the optimization objective does not capture user preferences precisely. Even for single-solution non-convex optimization, typical methods look for multiple local optima from random initial guesses before picking the best local optimum. Additionally, it is often desirable to obtain solutions to a family of optimization problems with parameters not known in advance, for instance, the weight of a regularization term, without having to restart from scratch.

Formally, we define a *multi-solution optimization* (MSO) problem to be the minimization $\min_{x \in \mathcal{X}} f_\tau(x)$, where $\tau \in \mathcal{T}$ encodes parameters of the problem, $\mathcal{X}$ is the search space of the variable $x$, and $f_\tau : \mathbf{R}^d \to \mathbf{R}$ is the objective function depending on $\tau$. The goal of MSO is to identify multiple solutions for each $\tau \in \mathcal{T}$, i.e., the set $\{x^* \in \mathcal{X} : f_\tau(x^*) = \min_{x \in \mathcal{X}} f_\tau(x)\}$, which can contain more than one element or even infinitely many elements. In this work, we assume that $\mathcal{X} \subset \mathbf{R}^d$ is bounded and that $d$ is small, and that $\mathcal{T}$ is, in a loose sense, a continuous space, such that the objective $f_\tau$ changes continuously as $\tau$ varies. To make gradient-based methods viable, we further assume that each $f_\tau$ is differentiable almost everywhere. As finding all global minima in the general case is extremely challenging, realistically our goal is to find a diverse set of local minima.

As a concrete example, for object detection, $\mathcal{T}$ could parameterize the space of images and $\mathcal{X}$ could be the 4-dimensional space of bounding boxes (ignoring class labels). Then, $f_\tau(x)$ could be the minimum distance between the bounding box $x \in \mathcal{X}$ and any ground truth box for image $\tau \in \mathcal{T}$. Minimizing $f_\tau(x)$ would yield *all* object bounding boxes for image $\tau$. Object detection can then be cast as solving this MSO on a training set of images and extrapolating to unseen images (Section 5.5). Object detection is a singular example of MSO where the ground truth annotation is widely available. In such cases, supervised learning can solve MSO by predicting a fixed number of solutions together with confidence scores using a set-based loss such as the Hausdorff distance. Unfortunately, such annotation is not available for most optimization problems in the wild where we only have access to the objective functions — this is the setting that our method aims to tackle.

Our work is inspired by the **proximal-point algorithm** (PPA), which applies the **proximal operator** of the objective function to an initial point iteratively to refine it to a local minimum. PPA is known to converge faster than gradient descent even when the proximal operator is approximated, both theoretically (Rockafellar, 1976; 2021) and empirically (e.g., Figure 2 of Hoheisel et al. (2020)). If the proximal operator of the objective function is available, then MSO can be solved efficiently by running PPA from a variety of initial points. However, obtaining a good approximation of the proximal operator for generic functions is difficult, and typically we have to solve a separate optimization problem for each evaluation of the proximal operator (Davis & Grimmer, 2019).

In this work, we approximate the proximal operator using a neural network that is trained using a straightforward loss term including only the objective and a *proximal term* that penalizes deviation from the input point. Crucially, our training does not require accessing the ground truth proximal operator. Additionally, neural parameterization allows us to learn the proximal operator for all $\{f_\tau\}_{\tau \in \mathcal{T}}$ by treating $\tau$ as an input to the network along with an application-specific encoder. Once trained, the learned proximal operator allows us to effortlessly run PPA from any initial point to arrive at a nearby local minimum; from a generative modeling point of view, the learned proximal operator implicitly encodes the solutions of an MSO problem as the pushforward of a prior distribution by iterated application of the operator. Such a formulation bypasses the need to predict a fixed number of solutions and can represent infinitely many solutions. The proximal term in our loss promotes the convexity of the formulation: applying recent results (Kawaguchi & Huang, 2019), we show that for weakly-convex objectives with Lipschitz gradients—in particular, objectives with bounded second derivatives—with practical degrees of over-parameterization, training converges globally and the ground truth proximal operator is recovered (Theorem 3.1 below). Such a global convergence result is not known for any previous learning-to-optimize method (Chen et al., 2021).

Literature on MSO is scarce, so we build a benchmark with a wide variety of applications including level set sampling, non-convex sparse recovery, max-cut, 3D symmetry detection, and object detection in images. When evaluated on this benchmark, our learned proximal operator reliably produces high-quality results compared to reasonable alternatives, while converging in a few iterations.

## 2 RELATED WORKS

**Learning to optimize.** Learning-to-optimize (L2O) methods utllize past optimization experience to optimize future problems more effectively; see (Chen et al., 2021) for a survey. *Model-free* L2O uses recurrent neural networks to discover new optimizers suitable for similar problems (Andrychowicz et al., 2016; Li & Malik, 2016; Chen et al., 2017; Cao et al., 2019); while shown to be practical, these methods have almost no theoretical guarantee for the training to converge (Chen et al., 2021). In comparison, we learn a problem-dependent proximal operator so that at test time we do not need access to objective functions or their gradients, which can be costly to evaluate (e.g. symmetry detection in Section 5.4) or unavailable (e.g. object detection in Section 5.5). *Model-based* L2O substitutes components of a specialized optimization framework or schematically unrolls an optimization procedure with neural networks. Related to proximal methods, Gregor & LeCun (2010) emulate a few iterations of proximal gradient descent using neural networks for sparse recovery with an $\ell^1$ regularizer, extended to non-convex regularizers by Yang et al. (2020); a similar technique is applied to susceptibility-tensor imaging in Fang et al. (2022). Gilton et al. (2021) propose a deep equilibrium model with proximal gradient descent for inverse problems in imaging that circumvents expensive backpropagation of unrolling iterations. Meinhardt et al. (2017) use a fixed denoising neural network as a surrogate proximal operator for inverse imaging problems. All these works use

schematics of proximal methods to design a neural network that is then trained with strong supervision. In contrast, we learn the proximal operator directly, requiring only access to the objectives; we do not need ground truth for inverse problems during training.

Existing L2O methods are not designed to recover multiple solutions: without a proximal term like in (2), the learned operator can degenerate even with multiple starts (Appendix D.3).

**Finding multiple solutions.** Many heuristic methods have been proposed to discover multiple solutions including niching (Brits et al., 2007; Li, 2009), parallel multi-starts (Larson & Wild, 2018), and deflation (Papadopoulos et al., 2021). However, all these methods do not generalize to similar but unseen problems.

Predicting multiple solutions at test time is universal in deep learning tasks like multi-label classification (Tsoumakas & Katakis, 2007) and detection (Liu et al., 2020). The typical solution is to ask the network to predict a fixed number of candidates along with confidence scores to indicate how likely each candidate is a solution (Ren et al., 2015; Li et al., 2019; Carion et al., 2020). Then the solutions will be chosen from the candidates using heuristics such as non-maximum suppression (Neubeck & Van Gool, 2006). Models that output a fixed number of solutions without taking into account the unordered set structure can suffer from "discontinuity" issues: a small change in set space requires a large change in the neural network outputs (Zhang et al., 2019). Furthermore, this approach cannot handle the case when the solution set is continuous.

**Wasserstein gradient flow.** Our formulation (2) corresponds to one step of JKO discretization of the Wasserstein gradient flow where the energy functional is the the linear functional dual to the MSO objective function (Jordan et al., 1998; Benamou et al., 2016). See the details in Appendix E. Compared to recent works on neural Wasserstein gradient flows (Mokrov et al., 2021; Hwang et al., 2021; Bunne et al., 2022), where a separate network parameterizes the pushforward map for every JKO step, our functional's linearity makes the pushforward map identical for each step, allowing end-to-end training using a single neural network. We additionally let the network input a parameter $\tau$, in effect learning a continuous family of JKO-discretized gradient flows.

## 3 METHOD

### 3.1 PRELIMINARIES

Given the objective $f_\tau : \mathbf{R}^d \to \mathbf{R}$ of an MSO problem parameterized by $\tau$, the corresponding **proximal operator** (Moreau, 1962; Rockafellar, 1976; Parikh & Boyd, 2014) is defined, for a fixed $\lambda \in \mathbf{R}_{>0}$, as

$$\text{prox}(x; \tau) := \arg\min_y \left\{ f_\tau(y) + \frac{\lambda}{2}\|y - x\|_2^2 \right\}. \tag{1}$$

The weight $\lambda$ in the **proximal term** $\lambda/2\|y - x\|_2^2$[1] controls how close $\text{prox}(x; \tau)$ is to $x$: increasing $\lambda$ will reduce $\|\text{prox}(x; \tau) - x\|_2$. For the $\arg\min$ in (1) to be unique, a sufficient condition is that $f_\tau$ is $\xi$-weakly convex with $\xi < \lambda$, so that $f_\tau(y) + \frac{\lambda}{2}\|y - x\|^2$ is strongly convex. The class of weakly convex functions is deceivingly broad: for instance, any twice differentiable function with bounded second derivatives (e.g. any $C^2$ function on a compact set) is weakly convex. When the function is convex, $\text{prox}(x; \tau)$ is precisely one step of the backward Euler discretization of integrating the vector field $-\nabla f_\tau$ with time step $1/\lambda$ (see Section 4.1.1 of Parikh & Boyd (2014)).

The **proximal-point algorithm** (PPA) for finding a local minimum of $f_\tau$ iterates

$$x^k := \text{prox}(x^{k-1}; \tau), \forall k \in \mathbf{N}_{\geq 1},$$

with initial point $x^0$ (Rockafellar, 1976). In practice, $\text{prox}(x; \tau)$ often can only be approximated, resulting in *inexact* PPA. When the objective function is locally indistinguishable from a convex function and $x^0$ is sufficiently close to the set of local minima, then with reasonable stopping criterion, inexact PPA converges linearly to a local minimum of the objective: the smaller $\lambda$ is, the faster the convergence rate becomes (Theorem 2.1-2.3 of Rockafellar (2021)).

---

[1]The usual convention is to use the reciprocal of $\lambda$ in front of the proximal term. We use a different convention to associate $\lambda$ with the convexity of (1).

## 3.2 Learning Proximal Operators

The fast convergence rate of PPA makes it a strong candidate for MSO: to obtain a diverse set of solutions for any $\tau \in \mathcal{T}$, we only need to run a few iterations of PPA from random initial points. The proximal term penalizes big jumps and prevents points from collapsing to a single solution. However, running a subroutine to approximate $\text{prox}(x; \tau)$ for every pair $(x, \tau)$ can be costly.

To overcome this issue, we *learn* the operator $\text{prox}(\cdot; \cdot)$ given access to $\{f_\tau\}_{\tau \in \mathcal{T}}$. A naïve way to learn $\text{prox}(\cdot; \cdot)$ is to first solve (1) to produce ground truth for a large number of $(x, \tau)$ pairs independently using gradient-based methods and then learn the operator using mean-squared error loss. However, this approach is costly as the space $\mathcal{X} \times \mathcal{T}$ can be large. Moreover, this procedure requires a stopping criterion for the minimization in (1), which is hard to design *a priori*.

Instead, we formulate the following end-to-end optimization over the space of functions:

$$\min_{\Phi: \mathcal{X} \times \mathcal{T} \to \mathcal{X}} \mathbf{E}_{\substack{x \sim \mu \\ \tau \sim \nu}} \left[ f_\tau(\Phi(x, \tau)) + \frac{\lambda}{2} \|\Phi(x, \tau) - x\|_2^2 \right], \tag{2}$$

where $x$ is sampled from $\mu$, a distribution on $\mathcal{X}$, and $\tau$ is sampled from $\nu$, a distribution on $\mathcal{T}$. To get (2) from (1), we essentially substitute $y$ with the output $\Phi(x, \tau)$ and integrate over the product probability distribution $\mu \otimes \nu$.

To solve (2), we parameterize $\Phi : \mathcal{X} \times \mathcal{T} \to \mathcal{X}$ using a neural network with additive and multiplicative residual connections (Appendix B). Intuitively, the implicit regularization of neural networks aligns well with the regularity of $\text{prox}(\cdot; \cdot)$: for a fixed $\tau$ the proximal operator $\text{prox}(\cdot; \tau)$ is 1-Lipschitz in local regions where $f_\tau$ is convex, while as the parameter $\tau$ varies $f_\tau$ changes continuously so $\text{prox}(x; \tau)$ should not change too much. To make (2) computationally practical during training, we realize $\nu$ as a training dataset. For the choice of $\mu$, we employ an importance sampling technique from Wang & Solomon (2019) as opposed to using $\text{unif}(\mathcal{X})$, the uniform distribution over $\mathcal{X}$, so that the learned operator can refine near-optimal points (Appendix C). To train $\Phi$, we sample a mini-batch of $(x, \tau)$ to evaluate the expectation and optimize using Adam (Kingma & Ba, 2014). For problems where the space $\mathcal{T}$ is structured (e.g. images or point clouds), we first embed $\tau$ into a Euclidean feature space through an encoder before passing it to $\Phi$. Such encoder is trained together with operator network $\Phi$. This allows us to use efficient domain-specific encoder (e.g. convolutional networks) to facilitate generalization to unseen $\tau$.

To extract multiple solutions at test time for a problem with parameter $\tau$, we sample a batch of $x$'s from $\text{unif}(\mathcal{X})$ and apply the learned $\Phi(\cdot, \tau)$ to the batch of samples a few times. Each application of $\Phi$ approximates a single step of PPA. From a distributional perspective, for $k \in \mathbf{N}_{\geq 0}$, we can view $\Phi^k$—the operator $\Phi$ applied $k$ times—as a generative model so that the pushforward distribution, $(\Phi^k)_\#(\text{unif}(\mathcal{X}))$, concentrates on the set of local minima approximates as $k$ increases. An advantage of our representation is that it can represent arbitrary number of solutions even when the set of minima is continuous (Figure 2). This procedure differs from those in existing L2O methods (Chen et al., 2021): at test time, we do not need access to $\{f_\tau\}_{\tau \in \mathcal{T}}$ or their gradients, which can be costly to evaluate or unavailable; instead we only need $\tau$ (e.g. in the case of object detection, $\tau$ is an image).

## 3.3 Convergence of Training

We have turned the problem of finding multiple solutions for each $f_\tau$ in the space $\mathcal{X}$ into the problem of finding a single solution for (2) in the space of functions. If the $f_\tau$'s are $\xi$-weakly convex with $\xi < \lambda$ and $\mu, \nu$ have full support, then the $\arg\min$ in (1) is unique for every pair $(x, \tau)$ and hence the functional solution of (2) is the unique proxmal operator $\text{prox}(\cdot; \tau)$.

If in addition the gradients of the objectives are Lipschitz, using recent learning theory results (Kawaguchi & Huang, 2019) we can show that with practical degrees of over-parameterization, gradient descent on neural network parameters of $\Phi$ converges globally during training. Suppose our training dataset is $S = \{(x_i, \tau_i)\}_{i=1}^n \subset \mathcal{X} \times \mathcal{T}$. Define the training loss, a discretized version of (2) using $S$, to be, for $g : \mathcal{X} \times \mathcal{T} \to \mathcal{X}$,

$$L(g) := \frac{1}{n} \sum_{i=1}^n \left[ f_{\tau_i}(g(x_i, \tau_i)) + \frac{\lambda}{2} \|g(x_i, \tau_i) - x_i\|_2^2 \right]. \tag{3}$$

**Theorem 3.1** (informal). *Suppose for any $\tau \in \mathcal{T}$, the objective $f_\tau$ is differentiable, $\xi$-weakly convex, and $\nabla f_\tau$ is $\zeta$-Lipschitz with $\xi \leq \lambda$. Then for any feed-forward neural network with $\tilde{\Omega}(n)$ total parameters[2] and common activation units, when the initial weights are drawn from a Gaussian distribution, with high probability, gradient descent on its weights using a fixed learning rate will eventually reach the minimum loss $\min_{g:\mathcal{X} \times \mathcal{T} \to \mathcal{X}} L(g)$. The number of iterations needed to achieve $\epsilon > 0$ training error is $O((\lambda + \zeta)/\epsilon)$, and when this occurs, if $\xi < \lambda$, then the mean-squared error of the learned proximal operator compared to the true one is $O(^{2\epsilon}/(\lambda - \xi))$ on training data.*

We state and prove Theorem 3.1 formally in Appendix A. Even though the optimization over network weights is non-convex, training can still result in a globally minimal loss and the true proximal operator can be recovered. In Appendix D.2, we empirically verify that when the objective is the $\ell^1$ norm, the trained operator converges to the true proximal operator, the shrinkage operator. In Appendix D.3, we study the effect of $\lambda$ in relation to the weakly-convex constant $\xi$ for the 2D cosine problem and compare to an L2O particle-swarm method (Cao et al., 2019).

We note a few gaps between Theorem 3.1 and our implementation. First, we use SGD with mini-batching instead of gradient descent. Second, instead of feed-forward networks, we use a network architecture with residual connections (Figure B.1), which works better empirically. Under these conditions, global convergence results can still be obtained, e.g., via (Allen-Zhu et al., 2019, Theorems 6 and 8), but with large polynomial bounds in $n$, $H$ for the network parameters. Another gap is caused by the restriction of the function class of the objectives. In several applications in Section 5, the objective functions are not weakly convex or have Lipschitz gradients, or we deliberately choose small $\lambda$ for faster PPA convergence; we empirically demonstrate that our method remains effective.

## 4 PERFORMANCE MEASURES

**Metrics.** Designing a single-valued metric for MSO is challenging since one needs to consider the diversity of the solutions as well each solution's level of optimality. For an MSO problem with parameter $\tau$ and objective $f_\tau$, the output of an MSO algorithm can be represented as a (possibly infinite) set of solutions $\{x_\alpha\}_\alpha \subset \mathcal{X}$ with objective values $u_\alpha := f_\tau(x_\alpha)$. Suppose we have access to ground truth solutions $\{y_\beta\}_\beta \subset \mathcal{X}$ with $v_\beta := f_\tau(y_\beta)$. Pick a threshold $t \in \mathbf{R}$ and denote $A_t := \{x_\alpha : u_\alpha \leq t\}, B_t := \{y_\beta : v_\beta \leq t\}$. Let $W$ be a random variable that is uniformly distributed on $\mathcal{X}$. Define a random variable

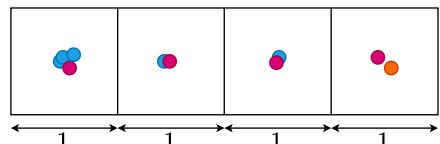

Figure 1: Interpretation of $D_t$. In this example, the witness $W$ is drawn uniformly from the union of four squares. If $A_t$ (resp. $B_t$) is the set of red (resp. blue) points, then $\mathbf{P}(D_t \approx 0) = 3/4$ and $\mathbf{P}(D_t \approx 0.5) = 1/4$, since $D_t$ is only non-zero when $W$ is in the rightmost square. This aligns well with the intuition that $3/4$ of the red points match with the blue ones. In comparison, the Hausdorff distance between $A_t$ and $B_t$ is approximately 1, which is the same as the Hausdorff distance between the orange point and $B_t$, despite the fact most of red points are close to the blue ones.

$$D_t := \frac{1}{2}\|\pi_{A_t}(W) - \pi_{B_t}(\pi_{A_t}(W))\|_2 + \frac{1}{2}\|\pi_{B_t}(W) - \pi_{A_t}(\pi_{B_t}(W))\|_2, \tag{4}$$

where $\pi_S(x) := \arg\min_{s \in S} \|x - s\|_2$. We call $W$ a *witness* of $D_t$, as it witnesses how different $A_t$ and $B_t$ are near $W$. To summarize the law of $D_t$, we define the *witnessed divergence* and *witnessed precision at* $\delta > 0$ as

$$\mathrm{WD}_t := \mathbf{E}[D_t] \quad \text{and} \quad \mathrm{WP}_t^\delta := \mathbf{P}(D_t < \delta). \tag{5}$$

Witnesses help handle unbalanced clusters that can appear in the solution sets. These metrics are agnostic to duplicates, unlike the chamfer distance or optimal transport metrics. Compared to alternatives like the Hausdorff distance, $\mathrm{WD}_t$ remains low if a small portion of $A_t, B_t$ are mismatched. We illustrate these metrics in Figure 1. One can interpret $\mathrm{WD}_t$ as a weighted chamfer distance whose weight is proportional to the volume of the $\ell^2$-Voronoi cell at each point in either set.

**Particle Descent: Ground Truth Generation.** A naïve method for MSO is to run gradient descent until convergence on randomly sampled particles in $\mathcal{X}$ for every $\tau \in \mathcal{T}$. We use this method to generate approximated ground truth solutions to compute the metrics in (5) when the ground truth is not available. This method is not directly comparable to ours since it cannot generalize to unseen

---

[2]We use $\tilde{\Omega}$ notation in the standard way, i.e., $f \in \tilde{\Omega}(n) \iff \exists k \in \mathbf{N}_{\geq 0}$ such that $f \in \Omega(n \log^k n)$.

$\tau$'s at test time. Remarkably, for highly non-convex objectives, particle descent can produce worse solutions than the ones obtained using the learned proximal operator (Figure D.7).

**Learning Gradient Descent Operators.** As there is no readily-available application-agnostic baseline for MSO, we propose the following method that learns iterations of the gradient descent operator. Fix $Q \in \mathbf{N}_{\geq 1}$ and a step size $\eta > 0$. We optimize an operator $\Psi$ via

$$\min_{\Psi: \mathcal{X} \times \mathcal{T} \to \mathcal{X}} \mathbf{E}_{\substack{x \sim \mu \\ \tau \sim \nu}} \big\| \Psi(x, \tau) - \Psi_Q^*(x; \tau) \big\|_2^2, \tag{6}$$

where $\Psi_Q^*(x; \tau)$ is the result of $Q$ steps of gradient descent on $f_\tau$ starting at $x$, i.e., $\Psi_0^*(x; \tau) = x$, and $\Psi_k^*(x; \tau) = \Psi_{k-1}^*(x; \tau) - \eta \nabla f_\tau(\Psi_{k-1}^*(x; \tau))$. Each iteration of minimizing (6) requires $Q$ evaluations of $\nabla f_\tau$, which can be costly (e.g., for symmetry detection in Section 5.4). We use importance sampling similar to Appendix C. An ODE interpretation is that $\Psi$ performs $Q$ iterations of *forward* Euler on the gradient field $\nabla f_\tau$, whereas the learned proximal operator performs a single iteration of *backward* Euler. We choose $Q = 10$ for all experiments except for symmetry detection (Section 5.4) where we choose $Q = 1$ because otherwise the training will take $> 200$ hours. As we will see in Figure D.6, aside from slower training, this approach struggles with non-smooth objectives due to the fixed step size $\eta$, while the learned proximal operator has no such issues.

## 5 APPLICATIONS

We consider five applications to benchmark our MSO method, chosen to highlight the ubiquity of MSO in diverse settings. We abbreviate POL for proximal operator learning (proposed method), GOL for gradient operator learning (Section 4), and PD for particle descent (Section 4). Further details about each application can be found in Appendix D. The source code for all experiments can be found at https://github.com/lingxiaoli94/POL.

### 5.1 SAMPLING FROM LEVEL SETS

**Formulation.** Level sets provide a concise and resolution-free implicit shape representation (Museth et al., 2002; Park et al., 2019; Sitzmann et al., 2020). Yet they are less intuitive to work with, even for straightforward tasks on discretized domains (meshes, point clouds) like visualizing or integration on the domain. We present an MSO formulation to sample from level sets, enabling the adaptation of downstream tasks to level sets.

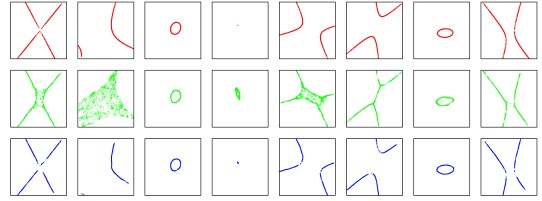

Figure 2: Visualization of the solutions for the conic section problem. Red, green, and blue indicate the solutions by PD, GOL, and POL respectively. See Figure D.3 for more examples.

Given a family of functions $\{g_\tau : \mathcal{X} \to \mathbf{R}^q\}_{\tau \in \mathcal{T}}$, for each $\tau$ suppose we want to sample from the 0-level set $g_\tau^{-1}(0)$. We formulate an MSO problem with objective $f_\tau(x) := \|g_\tau(x)\|_2^2$, whose global optima are precisely $g_\tau^{-1}(0)$. We do not need assumptions on level set topology or that the implicit function represents a distance field, unlike most existing methods (Park et al., 2019; Deng et al., 2020; Chen et al., 2020).

**Benchmark.** We consider sampling from conic sections. We keep this experiment simple so as to visualize the solutions easily. Let $\mathcal{X} = [-5, 5]^2$ and $\mathcal{T} = [-1, 1]^6$. For $\tau = (A, B, C, D, E, F) \in \mathcal{T}$, define $g_\tau$ to be $g_\tau(x_1, x_2) := Ax^2 + Bxy + Cy^2 + Dx + Ey + F$. Since $f_\tau = (g_\tau)^2$ is a defined on a compact $\mathcal{X}$, it satisfies the conditions of Theorem 3.1 for a large $\lambda$, but a large $\lambda$ corresponds to small PPA step size. Empirically, small $\lambda$ for POL gave decent results compared to GOL: Figure 2 illustrates that POL consistently produces sharper level sets for both hyperbolas ($B^2 - 4AC > 0$) and ellipses ($B^2 - 4AC < 0$). Figure D.4 shows that POL yields significantly higher $\text{WP}_t^\delta$ than GOL for small $\delta$, implying that details are well recovered. Figure D.5 verifies that iterating the trained operator of POL converges much faster than that of GOL. It is straightforward to extend this setting to sample from more complicated implicit shapes parameterized by $\tau$.

### 5.2 SPARSE RECOVERY

**Formulation.** In signal processing, the *sparse recovery* problem aims to recover a signal $x^* \in \mathcal{X} \subset \mathbf{R}^d$ from a noisy measurement $y \in \mathbf{R}^m$ distributed according to $y = Ax^* + e$, where $A \in \mathbf{R}^{m \times d}$,

$m < d$, and $e$ is measurement noise (Beck & Teboulle, 2009). In applications like imaging and speech recognition, the signals are *sparse*, with few non-zero entries (Marques et al., 2018). Hence, the goal of sparse recovery is to recover a sparse $x^*$ given $A$ and $y$.

A common way to encourage sparsity is to solve least-squares plus an $\ell^p$ norm on the signal:

$$\min_{x \in \mathcal{X}} \|Ax - y\|_2^2 + \alpha \|x\|_p^p, \tag{7}$$

for $\alpha, p > 0$ and $\|x\|_p^p := \sum_{i=1}^d (x_i^2 + \epsilon)^{p/2}$ for a small $\epsilon$ to prevent instability. We consider the non-convex case where $0 < p < 1$. Compared to convex alternatives like in LASSO ($p = 1$), non-convex $\ell^p$ norms require milder conditions under which the global optima of (7) are the desired sparse $x^*$ (Chartrand & Staneva, 2008; Chen & Gu, 2014).

To apply our MSO framework, we define $\tau = (\alpha, p) \in \mathcal{T}$ and $f_\tau$ to be the objective (7) with corresponding $\alpha, p$. Compared to existing methods for non-convex sparse recovery (Lai et al., 2013), our method can recover multiple solutions from the non-convex landscape for a family of $\alpha$'s and $p$'s without having to restart. The user can adjust parameters $\alpha, p$ to quickly generate candidate solutions before choosing a solution based on their preference.

**Benchmark.** Let $\mathcal{X} = [-2, 2]^8$, $\mathcal{T} = [0, 1] \times [0.2, 0.5]$. We consider highly non-convex $\ell^p$ norms with $p \in [0.2, 0.5]$ to test our method's limits. We choose $d = 8$ and $m = 4$, and sample the sparse signal $x^*$ uniformly in $\mathcal{X}$ with half of the coordinates set to 0. We then sample entries in $A$ i.i.d. from $\mathcal{N}(0, 1)$ and generate $y = Ax^* + e$ where $e \sim \mathcal{N}(0, 0.1)$. Although $\|x\|_p^p$ is not weakly convex, POL achieves decent results (Figure D.6). Notably, POL often reaches a better objective than PD (Figure D.7) while retaining diversity, even though POL uses a much bigger step size ($1/\lambda = 0.1$ compared to PD's $10^{-5}$) and needs to learn a different operator for an entire family of $\tau \in \mathcal{T}$. In Figure D.8, we additionally compare POL with proximal gradient descent (Tibshirani et al., 2010) for $p = 1/2$ where the corresponding thresholding formula has a closed-form (Cao et al., 2013). Remarkably, we have observed superior performance of POL against such a strong baseline.

## 5.3 RANK-2 RELAXATION OF MAX-CUT

**Formulation.** MSO can be applied to solve combinatorial problems that admit smooth non-convex relaxations. Here, we consider the classical problem of finding the maximum cut of an undirected graph $G = (V, E)$, where $V = \{1, \ldots, n\}$, $E \subset V \times V$, with edge weights $\{w_{ij}\} \subset \mathbf{R}$ so that $w_{ij} = 0$ if $(i, j) \notin E$. The goal is to find $\{x_i\} \in \{-1, +1\}^V$ to maximize $\sum_{i,j} w_{ij}(1 - x_i x_j)$.

Burer et al. (2002) propose solving $\min_{\theta \in \mathbf{R}^n} \sum_{i,j} w_{ij} \cos(\theta_i - \theta_j)$, a rank-2 non-convex relaxation of the max-cut problem. This objective inherits weak convexity from cosine, so it satisfies the conditions of Theorem 3.1. In practice, instead of using angles as the variables which are ambiguous up to $2\pi$, we represent each variable as a point on the unit circle $S^1$, so we choose $\mathcal{X} = (S^1)^n$ and $\mathcal{T}$ be the space of all edge weights with $n$ vertices. For $\tau = \{\tau_{ij}\} \in \mathcal{T}$ corresponding to a graph with edge weights $\{\tau_{ij}\}$, we define, for $x \in \mathcal{X}$,

$$f_\tau(x) := \sum_{i,j} \tau_{ij} x_i^\top x_j. \tag{8}$$

After minimizing $f_\tau$, we can find cuts using a Goemans & Williamson-type procedure (1995). Instead of using heuristics to find optima near a solution (Burer et al., 2002), our method can help the user effortlessly explore the set of near-optimal solutions without hand-designed heuristics.

**Benchmark.** We apply our formulation to $K_8$, the complete graph with 8 vertices. Hence $\mathcal{X} = (S^1)^8 \subset \mathbf{R}^{16}$. We choose $\mathcal{T} = [0, 1]^{28}$ as there are 28 edges in $K_8$. We mix two types of random graphs with 8 vertices in training and testing: Erdős-Rényi graphs with $p = 0.5$ and $K_8$ with uniform edge weights in $[0, 1]$. Figure 3 shows that POL can generate diverse set of max cuts. Quantitatively, compared to GOL, POL achieves better witnessed metrics (Figure D.10).

## 5.4 SYMMETRY DETECTION OF 3D SHAPES

**Formulation.** Geometric symmetries are omnipresent in natural and man-made objects. Knowing symmetries can benefit downstream tasks in geometry and vision (Mitra et al., 2013; Shi et al., 2020; Zhou et al., 2021). We consider the problem of finding all reflection symmetries of a 3D surface.

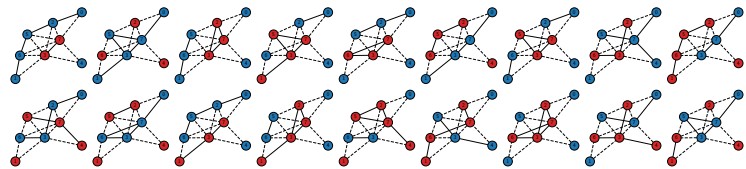

Figure 3: 18 different max cuts (max cut value 10) of a graph generated by our method. Red and blue vertices indicate the two vertex set separated by the cut. Vertex 0 is set to blue to remove the duplicates obtained by swapping the colors. See Figure D.12 for more results.

Let $\tau$ be a shape representation (e.g. point cloud, multi-view scan), and let $\mathcal{M}_\tau \subset \mathbf{R}^3$ denote the corresponding triangular mesh that is available for the training set. As reflections are determined by the reflectional plane, we set $\mathcal{X} = S^2 \times \mathbf{R}_{\geq 0}$, where $x = (n, d) \in \mathcal{X}$ denotes the plane with unit normal $n \in S^2 \subset \mathbf{R}^3$ and intercept $d \in \mathbf{R}_{\geq 0}$ (we assume $d \geq 0$ to remove the ambiguity of $(-n, -d)$ representing the same plane). Let $R_x : \mathbf{R}^3 \to \mathbf{R}^3$ denote the corresponding reflection. Perfect symmetries of $\mathcal{M}_\tau$ satisfy $R_x(\mathcal{M}_\tau) = \mathcal{M}_\tau$. Let $s_\tau : \mathbf{R}^3 \to \mathbf{R}$ be the (unsigned) distance field of $\mathcal{M}_\tau$ given by $s_\tau(p) = \min_{q \in \mathcal{M}_\tau} \|p - q\|_2$. Inspired by Podolak et al. (2006), we define the MSO objective to be

$$f_\tau(x) := \mathbf{E}_{p \sim \mathcal{M}_\tau}[s_\tau(R_x(p))], \tag{9}$$

where a batch of $p$ is sampled uniformly from $\mathcal{M}_\tau$ when evaluating the expectation. Although $f_\tau$ is stochastic, since we use point-to-mesh distances to compute $s_\tau$, perfect symmetries will make (9) zero with probability one. Compared to existing methods that either require ground truth symmetries obtained by human annotators (Shi et al., 2020) or detect only a small number of symmetries (Gao et al., 2020), our method applied to (9) finds arbitrary numbers of symmetries including continuous ones and can generalize to unseen shapes, without needing ground truth symmetries as supervision.

**Benchmark.** We detect reflection symmetries for mechanical parts in the MCB dataset (Kim et al., 2020). We choose $\mathcal{T}$ to be the space of 3D point clouds representing mechanical parts. From the mesh of each shape, we sample 2048 points with their normals uniformly and use DGCNN (Wang et al., 2019) to encode the oriented point clouds. Figure 4 show our method's results on a selection of models in the test dataset; for per-iteration PPA results of our method, see Figure D.13. Figure D.11 shows that POL achieves much higher witnessed precision compared to GOL.

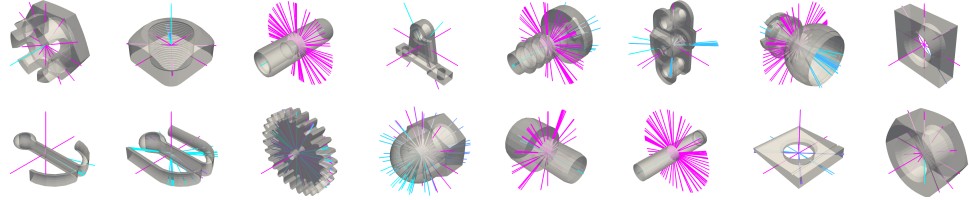

Figure 4: Symmetry detection results. Each reflection is represented as a colored line segment representing the normal of the reflection plane with one endpoint on the plane. Pink indicates better objective values, while blue indicates worse. Our method is capable of detecting complicated discrete symmetries as well as continuous families of cylindrical reflectional symmetries.

## 5.5 OBJECT DETECTION IN IMAGES

**Formulation.** Identifying objects in an image is a central problem in vision on which recent works have made significant progress (Ren et al., 2015; Carion et al., 2020; Liu et al., 2021). We consider a simplified task where we drop the class labels and predict only bounding boxes. Let $b = (x, y, w, h) \in \mathcal{X} = [0, 1]^4$ denote a box with (normalized) center coordinates $(x, y)$, width $w$, and height $h$. We choose $\mathcal{T}$ to be the space of images. Suppose an image $\tau$ has $K_\tau$ ground truth object bounding boxes $\{b_i^\tau\}_{i=1}^{K_\tau}$. We define the MSO objective to be $f_\tau(x) := \min_{i=1}^{K_\tau} \|b_i^\tau - x\|_1$; its minimizers are exactly $\{b_i^\tau\}_{i=1}^{K_\tau}$. Although the objective may seem trivial, its gradients reveal the $\ell^1$-Voronoi diagram formed by $b_i^\tau$'s when training the proximal operator. Different from existing approaches, we encode the distribution of bounding boxes conditioned on each image in the learned proximal operator without needing to predict confidence scores or a fixed number of boxes. A similar idea based on diffusion is recently proposed by Chen et al. (2022).

**Benchmark.** We apply the above MSO formulation to the COCO2017 dataset (Lin et al., 2014). As $\tau$ is an image, we fine-tune ResNet-50 (He et al., 2016) to encode $\tau$ into a vector $z$ that can be consumed by the operator network (Figure B.1).

Table 1: Object detection results. $\text{WD}_\infty$ (resp. $\text{WP}_\infty^{0.1}$) is the witnessed divergence (resp. precision) in (5) with $t = \infty$ (i.e. keeping all solutions), averaged over 10 trials (standard deviation $< 10^{-3}$). Precision and recall are computed with Hungarian matching as no confidence score is available for the usual greedy matching (see Appendix D.8). FRCNN$(.S)$ (Ren et al., 2015) means keeping predictions with confidence $\geq S\%$ for Faster R-CNN.

| METHOD | $\text{WD}_\infty$ | $\text{WP}_\infty^{0.1}$ | PRECISION | RECALL |
|---|---|---|---|---|
| FRCNN$(.80)$ | **0.140** | **0.624** | 0.778 | **0.650** |
| FRCNN$(.95)$ | 0.162 | 0.589 | **0.887** | 0.515 |
| FN | 0.161 | 0.481 | 0.139 | **0.577** |
| GOL | 0.251 | 0.243 | 0.508 | 0.282 |
| POL (OURS) | **0.149** | **0.590** | **0.817** | 0.442 |

In addition to GOL, we design a baseline method FN that uses the same ResNet-50 backbone and predicts a fixed number of boxes using the chamfer distance as the training loss. Table 1 compares the proposed methods with alternatives and the highly-optimized Faster R-CNN (Ren et al., 2015) on the test dataset. Since we do not output confidence scores, the metrics are computed solely based on the set of predicted boxes. Our method achieves significantly better results than FN and GOL. Compared to the Faster R-CNN, we achieve slightly worse results with $40.7\%$ fewer network parameters. While Faster R-CNN contains highly-specialized modules such as the regional proposal network, in our method we simply feed the image feature vector output by ResNet-50 to a general-purpose operator network. Incorporating specialized architectures like region proposal networks into our proximal operator learning framework for object detection is an exciting future direction. We visualize the effect of PPA using the learned proximal operator in Figure 5. Further qualitative results (Figure D.14) and details can be found in Appendix D.8.

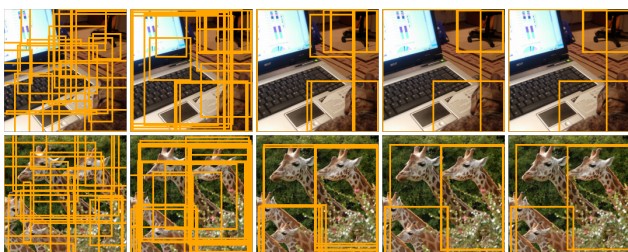

Figure 5: First 4 iterations of PPA using the learned proximal operator on 20 randomly initialized boxes (leftmost column). Only a few iterations are needed for the boxes to form distinctive clusters.

## 6  CONCLUSION

Our work provides a straightforward and effective method to learn the proximal operator of MSO problems with varying parameters. Iterating the learned operator on randomly initialized points efficiently yields multiple optima to the MSO problems. Beyond promising results on our benchmark tasks, we see many exciting future directions that will further improve our pipeline.

A current limitation is that at test time the optimal number of iterations to apply the learned operator is not known ahead of time (see end of Appendix D.1). One way to overcome this limitation would be to train another network that estimates when to stop. This measurement can be the objective itself if the optimum value is known *a priori* (e.g., sampling from level sets) or the gradient norm if objectives are smooth. One other future direction is to learn a proximal operator that adapts to multiple $\lambda$'s. This way, the user can easily experiment with different $\lambda$'s and to enable PPA with growing step sizes for super-linear convergence (Rockafellar, 1976; 2021). Another direction is to study how much we can relax the assumption that $\mathcal{X}$ is a low-dimensional Euclidean space. Our method could remain effective when $\mathcal{X}$ is a low-dimensional submanifold of a high-dimensional Euclidean space. The challenges would be to constrain the proximal operator to a submanifold and to design a proximal term that is more suitable than the ambient $\ell^2$ norm.

**Reproducibility statement.** The complete source code for all experiments can be found at `https://github.com/lingxiaoli94/POL`. Detailed instructions are given in `README.md`. We have further included a tutorial on how to extend the framework to custom problems—see "Extending to custom problems" section where we include a toy physics problem of finding all rest configurations of an elastic spring. For all our experiments, the important details are provided in the main text, while the remaining details needed to reproduce results exactly are included in the appendix.

**Acknowledgements** We thank Chenyang Yuan for suggesting the rank-2 relaxation of max-cut problems. The MIT Geometric Data Processing group acknowledges the generous support of Army Research Office grants W911NF2010168 and W911NF2110293, of Air Force Office of Scientific Research award FA9550-19-1-031, of National Science Foundation grants IIS-1838071 and CHS-1955697, from the CSAIL Systems that Learn program, from the MIT–IBM Watson AI Laboratory, from the Toyota–CSAIL Joint Research Center, from a gift from Adobe Systems, and from a Google Research Scholar award.

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

## A  CONVERGENCE OF TRAINING

We formally state and prove Theorem 3.1 via the following Proposition A.1 and Proposition A.2.

**Proposition A.1.** *Suppose*

1. $\mathcal{T} \subset \mathbf{R}^r$ *for some* $r \in \mathbf{N}_{\geq 1}$;

2. *for any* $\tau \in \mathcal{T}$, *the objective* $f_\tau$ *is differentiable,* $\xi$-*weakly convex, and* $\nabla f_\tau$ *is* $\zeta$-*Lipschitz, i.e.,*

$$\|\nabla f_\tau(x_1) - \nabla f_\tau(x_2)\|_2 \leq \zeta \|x_1 - x_2\|_2,$$

*with* $\xi \leq \lambda$.

3. *the activation function* $\sigma(x)$ *used is proper, real analytic, monotonically increasing and 1-Lipschitz, e.g., sigmoid, hyperbolic tangent.*

*For any* $\delta > 0$, $H \geq 2$, $n \in \mathbf{N}_{\geq 1}$, *assume* $\Phi$ *is an* $H$-*layer feed-forward neural network with hidden layer sizes* $m_1, \ldots, m_H$ *satisfying*

$$m_1, \ldots, m_{H-2} \geq \Omega(H^2 \log(Hn^2/\delta)),$$
$$m_{H-1} \geq \Omega(\log(Hn^2/\delta)), \quad m_H \geq \Omega(n).$$

*Let* $D$ *denote the total number of weights in* $\Phi$. *Then* $D = \tilde{\Omega}(n)$. *Moreover, there exists a learning rate* $\eta \in \mathbf{R}^D$ *such that for any dataset* $S = \{(x_i, \tau_i)\}_{i=1}^n$ *of size* $n$ *with the training loss* $L$ *defined as in* (3), *for any* $\epsilon > 0$, *with probability at least* $1 - \delta$ *(over random Gaussian initial weights* $\theta^0$ *of* $\Phi$), *there exists* $t = O(c_r(\lambda + \zeta)/\epsilon)$ *such that* $L(\Phi(\cdot, \cdot; \theta^t)) \leq L^* + \epsilon$, *where* $\|\theta^t\|_2^2$ *stays bounded,* $L^* := \min_{g \in \mathcal{X} \times \mathcal{T} \to \mathcal{X}} L(g)$ *is the global minimum of the functional* $L$, $(\theta^k)_{k \in \mathbf{N}}$ *is the sequence generated by gradient descent* $\theta^{k+1} := \theta^k - \eta \odot \nabla_\theta L(\Phi(\cdot, \cdot; \theta^k))$, *and* $c_r$ *depends only on* $L$ *and the initialization* $\theta^0$.

*Proof of Proposition A.1.* The theorem is an application of Theorem 1 in Kawaguchi & Huang (2019) with the following modifications.

For $i \in [n]$, define $\ell_i(x) := f_{\tau_i}(x) + \frac{\lambda}{2}\|x - x_i\|_2^2$. To check Assumption 1 of Kawaguchi & Huang (2019), observe

$$\nabla_x \ell_i(x) = \nabla f_{\tau_i}(x) + \lambda(x - x_i),$$
$$\nabla_x^2 \ell_i(x) = \nabla^2 f_{\tau_i}(x) + \lambda I_d.$$

Hence the assumption that $f_{\tau_i}$ is $\xi$-weakly convex implies that

$$\nabla^2 f_{\tau_i}(x) + \lambda I_d \succcurlyeq \nabla^2 f_{\tau_i}(x) + \xi I_d \succcurlyeq 0.$$

Hence $\ell_i$ is convex. The assumption that $\nabla f_{\tau_i}$ is $\zeta$-Lipschitz implies, for any $x_1, x_2 \in \mathcal{X} \times \mathcal{T}$,

$$\begin{aligned}
\|\nabla \ell_i(x_1) - \nabla \ell_i(x_2)\|_2 &= \|\nabla f_{\tau_i}(x_1) - \nabla f_{\tau_i}(x_2) + \lambda(x_2 - x_1)\|_2 \\
&\leq \|\nabla f_{\tau_i}(x_1) - \nabla f_{\tau_i}(x_2)\|_2 + \lambda\|x_1 - x_2\|_2 \\
&\leq (\lambda + \zeta)\|x_1 - x_2\|_2.
\end{aligned}$$

Hence $\nabla \ell_i$ is $(\lambda + \zeta)$-Lipschitz.

An input vector to the neural network $\Phi$ is the concatenation $(x, \tau) \in \mathbf{R}^{d+r}$. Kawaguchi & Huang (2019) assume that the input data points are normalized to have unit length. This is not an issue, as we can scale down $(x_i, \tau_i)$ uniformly to be contained in a unit ball, then pad $\tau_i$ one extra coordinate to make $\|(x_i, \tau_i)\|_2 = 1$ for all $i \in [n]$, similar to the argument given in the footnotes before Assumption 2.1 of Allen-Zhu et al. (2019).

Lastly, we mention explicitly lower bounds for the layer sizes that are used in the proof of Theorem 1 of Kawaguchi & Huang (2019) (see the paragraph below Lemma 3), instead of stating a single bound on the total number of weights in the statement of Theorem 1. This is because Theorem 1 only states that there *exists* a network of size $\tilde{\Omega}(n)$ for which training converges, whereas *every* network satisfying the layer-wise bounds will have the same convergence guarantee. □

Next we show that once the training loss is $\epsilon$ away from the global minimum, we can guarantee that the approximation error on the training data in the mean-squared sense is small: i.e., the learned operator $\Phi(\cdot, \cdot; \theta)$ is close to the true proximal operator (1).

**Proposition A.2.** *Suppose for any $\tau \in \mathcal{T}$, the objective $f_\tau$ is differentiable and $\xi$-weakly convex with $\xi < \lambda$, where $\lambda$ is the proximal regularization weight of the training loss $L(g)$ defined in (3). Let $\theta$ be the weight of the network $\Phi$ such that $L(\Phi(\cdot, \cdot; \theta)) \leq L^* + \epsilon$ where $L^* := \min_{g \in \mathcal{X} \times \mathcal{T} \to \mathcal{X}} L(g)$ is the global minimum of the functional $L$. Let $\mathrm{prox}(\cdot; \cdot)$ be the true proximal operator defined in (1). Then the mean-squared error on the training data is bounded by*

$$\frac{1}{n} \sum_{i=1}^{n} \|\Phi(x_i, \tau_i; \theta) - \mathrm{prox}(x_i; \tau_i)\|_2^2 \leq \frac{2\epsilon}{\lambda - \xi}. \tag{10}$$

*Proof.* Clearly $L^* = L(\mathrm{prox}(\cdot; \cdot))$, i.e., the minimum of $L$ is achieved with the true proximal operator. Define $h_i : \mathcal{X} \to \mathbf{R}$ by $h_i(x) := f_{\tau_i}(x) + \frac{\lambda}{2}\|x - x_i\|_2^2$, so that we can write $L(g) = \frac{1}{n} \sum_{i=1}^{n} h_i(g(x_i, \tau_i))$. By the assumption on weak convexity, each $h_i$ is $(\lambda - \xi)$-strongly convex. This implies for any $x, y \in \mathcal{X}$,

$$h_i(x) \geq h_i(y) + \nabla h_i(y)^\top (x - y) + \frac{\lambda - \xi}{2}\|x - y\|_2^2. \tag{11}$$

The minimum of $h_i$ is achieved at $\mathrm{prox}(x_i; \tau_i)$ by the definition of prox. Differentiability and convexity imply $\nabla h_i(\mathrm{prox}(x_i; \tau_i)) = 0$. Hence setting $y = \mathrm{prox}(x_i; \tau_i)$ in (11) implies, for any $x \in \mathcal{X}$,

$$h_i(x) - h_i(\mathrm{prox}(x_i; \tau_i)) \geq \frac{\lambda - \xi}{2}\|x - \mathrm{prox}(x_i; \tau_i)\|_2^2.$$

Now by the definition of (3),

$$\begin{aligned}
\epsilon &\geq L(\Phi(\cdot, \cdot; \theta)) - L^* = L(\Phi(\cdot, \cdot; \theta)) - L(\mathrm{prox}(\cdot, \cdot)) \\
&= \frac{1}{n} \sum_{i=1}^{n} [h_i(\Phi(x_i, \tau_i; \theta)) - h_i(\mathrm{prox}(x_i; \tau_i))] \\
&\geq \frac{1}{n} \sum_{i=1}^{n} \frac{\lambda - \xi}{2}\|\Phi(x_i, \tau_i; \theta) - \mathrm{prox}(x_i; \tau_i)\|_2^2.
\end{aligned}$$

Rearranging terms we obtain the desired result. $\square$

## B  NETWORK ARCHITECTURES

The network architecture we use to parameterize the operators for both POL and GOL is identical and is shown in Figure B.1. The encoder of $\tau$ will be chosen depending on the application. For our conic section (5.1), sparse recovery (5.2), and max-cut (5.3) benchmarks, the encoder is just the identity map. For symmetry detection (5.4), $\tau$ is a point cloud and we use DGCNN (Wang et al., 2019). For object detection (5.5), we use ResNet-50 (He et al., 2016). Inspired by Dinh et al. (2016), we include both additive and multiplicative coupling in the residual blocks. At the same time, since we do not need bijectivity of the operator (and proximal operators should not be) nor access to the determinant of the Jacobian, we do not restrict ourselves to a map with triangular structure as in Dinh et al. (2016). We use 3 residual blocks for all applications, except for symmetry detection where we use 5 blocks which give slightly improved performance.

Our architecture is economical: the model size (excluding the application-specific encoder) is under 2MB for all applications we consider. This also makes iterating the operators fast at test time. Note that the application-specific encoder only needs to be run once at each test time $\tau$ as the encoded vector $z$ can be reused (Figure B.1).

## C  IMPORTANCE SAMPLING VIA UNFOLDING PPA

Directly optimizing (2) or (6) using mini-batching may not yield an operator that can refine a near-optimal solution, if $\mu$ is taken to be $\mathrm{unif}(\mathcal{X})$, the uniform measure on $\mathcal{X}$ (more precisely, the $d$-dimensional Lebesgue measure restricted to $\mathcal{X}$ and normalized to a probability distribution). Instead,

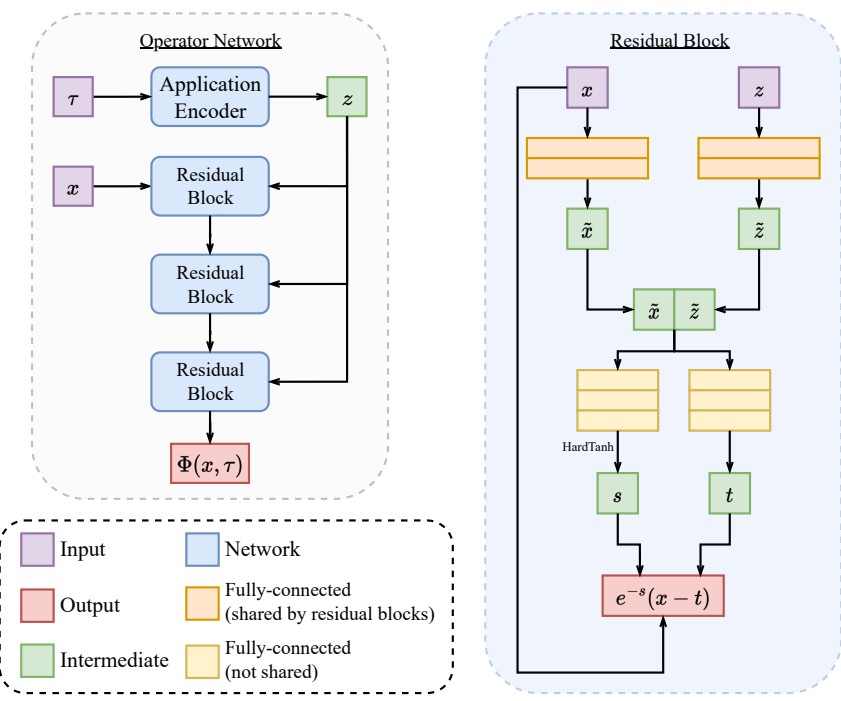

Figure B.1: Network architecture for the operators used in POL and GOL. We use ReLU as the activation after all intermediate linear layers, except for predicting the scaling in the residual block, where we use HardTanh to ensure $s \in [-2, 2]$. For the shared 2-layer fully-connected network, the hidden layer sizes are $256, 128$. For the 3-layer fully-connected network in each residual block, the hidden layer sizes are $128, 128, 128$.

we would like to sample from a distribution that puts more probability density on near-optimal solutions. We achieve this goal as follows, inspired by Wang & Solomon (2019). Let $\Phi$ denote the network with weights after $t$ training iterations. For $k \in \mathbf{N}_{\geq 0}$, denote $\mu^k := (\Phi^k)_{\#}(\text{unif}(\mathcal{X}))$. For a fixed $K \in \mathbf{N}_{\geq 1}$, we set $\mu := \frac{1}{K+1} \sum_{k=0}^{K} \mu^k$. Then, for training iteration $t + 1$, we optimize the objective (2) or (6) with the constructed $\mu$. Note this modification does not introduce any bias for POL (similarly for GOL), in the sense that the optimal solution to (2) is still the true proximal operator since $\mu$ has full support, yet it puts more density in near-optimal regions as $t$ increases. In practice, we choose $K = 5$ or $K = 10$. For the choice of other hyper-parameters, see Appendix D.1.

# D    DETAILED RESULTS

## D.1    HYPER-PARAMETERS

Unless mentioned otherwise, the following hyper-parameters are used.

In each training iteration of POL and GOL, we sample 32 problem parameters from the training dataset of $\mathcal{T}$, and 256 of $x$'s from $\text{unif}(\mathcal{X})$ when computing (2) or (6) using the importance sampling trick in Appendix C. The learning rate of the operator is kept at $10^{-4}$ for both POL and GOL, and by default we train the operator network for $2 \times 10^5$ iterations. This is sufficient for the loss to converge for both POL and GOL in most cases. Since GOL requires multiple evaluations of the gradient of the objective, it typically trains two or more times slower than POL. For the proximal weight $\lambda$ of POL, we choose it based on the scale of the objective and the dimension of $\mathcal{X}$; see Table D.1. All training is done on a single NVIDIA RTX 3090 GPU.

For the step size $\eta$ in GOL, we start with $1/\lambda$ (so same step size as POL in the forward/backward Euler sense) and then slowly increase it (so fewer iterations are needed for convergence) without

Table D.1: Choices of $\lambda$ for all applications considered. $\mathcal{X}$ is the search space of solutions, and $d$ is the dimension of the Euclidean space where we embed $\mathcal{X}$ (so it might be greater than the intrinsic dimension of $\mathcal{X}$).

| Application | $\mathcal{X}$ | $d$ | $\lambda$ |
|---|---|---|---|
| conic section (5.1) | $[-5, 5]^2$ | 2 | 0.1 |
| sparse recovery (5.2) | $[-2, 2]^8$ | 8 | 10.0 |
| max-cut (5.3) | $(S^1)^8$ | 16 | 10.0 |
| symmetry detection (5.4) | $S^2 \times \mathbf{R}_{>0}$ | 4 | 1.0 |
| object detection (5.5) | $[0, 1]^4$ | 4 | 1.0 |

degrading the metrics. When evaluating (6), we set $Q = 10$ in all experiments except for symmetry detection, where we use $Q = 1$ because otherwise the training will take $> 200$ hours. For PD, we choose a step size small enough so as to not miss significant minima and a sufficient number of iterations for the loss (i.e. the objectives) to fully converge.

For evaluation, the number of iterations to apply the trained operators is chosen to be enough so that the objective converges. This number will be chosen separately for each application and method. By default, 1024 solutions are extracted from each method, and 1024 witnesses are sampled to compute $\text{WD}_t$ and $\text{WP}_t^\delta$, averaged over test dataset and over 10 trials with standard deviation provided (in most cases the standard deviation is two orders of magnitude smaller than the metrics). We filter out solutions that do not lie in $\mathcal{X}$.

A limitation for both POL and GOL is that when the solution set is continuous, too many applications of the learned operator can cause the solutions to collapse. We suspect this is because even with the importance sampling trick (Appendix C), during training the operators may never see enough input that are near-optimal to learn the correct refinement needed to recover the continuous solution set. A future direction is to have another network to predict a confidence score for each $x \in \mathcal{X}$ so that at test time the user knows when to stop iterating the operator, e.g., when the objective value and its gradient are small enough; see the discussion in Section 6.

### D.2 CONVERGENCE TO THE PROXIMAL OPERATOR

To empirically verify Proposition A.2, that our method can faithfully approximate the true proximal operators of the objectives, we conduct the following simple experiments. We consider the function $f(x) = \|x\|_1$ for $x \in \mathcal{X} = [-1, 1]^d$ and treat $\mathcal{T}$ as a singleton. Its proximal operator $\text{prox}(x) = \arg\min_y \|y\|_1 + \|y - x\|_2^2$ is known in closed form as the shrinkage operation, defined coordinate-wise as:

$$\text{prox}(x)_i = \begin{cases} x_i - 1/2 & x_i \geq 1/2 \\ 0 & |x_i| \leq 1/2 \\ x_i + 1/2 & x_i \leq -1/2. \end{cases} \tag{12}$$

For each dimension of $d = 2, 4, 8, 16, 32$, we train an operator network $\Phi$ (Figure B.1) using (2) as the loss with learning rate $10^{-3}$. Figure D.1 shows the mean-squared-error $\|\Phi(x) - \Phi^*(x)\|_2^2$ scaled by $1/d$ and averaged over 1024 samples vs. the training iterations, where $\Phi^*$ is the shrinkage operation (12). We see that the trained operator indeed converges to $\Phi^*$ as predicted by Proposition A.2, and the convergence speed is faster in smaller dimensions.

### D.3 EFFECT OF THE PROXIMAL TERM

In this section, we study the necessity of the proximal term $\lambda/2 \|\Phi(x, \tau) - x\|^2$ in (2). Without such a term, the learned operator can degenerate. For example, consider (1) in (Chen et al., 2021), which minimizes $\min_\Phi \sum_{t=1}^T w_t f(x_t)$ with $x_{t+1} = x_t - \Phi(x_t, \nabla f(x_t), \ldots)$ for all $t$ (with adapted notation). Suppose $x^*$ is one global optimum of $f$ but is not the only one. Then $\Phi(x, \ldots) := x - x^*$ clearly minimizes the objective, yet the update steps will always set $x_t = x^*$ regardless of the initial positions.

To further illustrate the effect of different choices of $\lambda$, consider the 2D cosine function $f(x) = -\sum_{i=1}^2 10 \cos(2\pi x_i)$ for $x \in \mathcal{X} = [-5, 5]^2$ and a singleton $\mathcal{T}$. This function is $\xi$-weakly convex

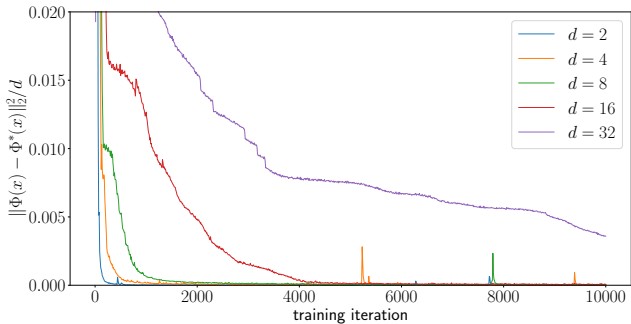

Figure D.1: Convergence to the true proximal operator of $f(x) = \|x\|_1$.

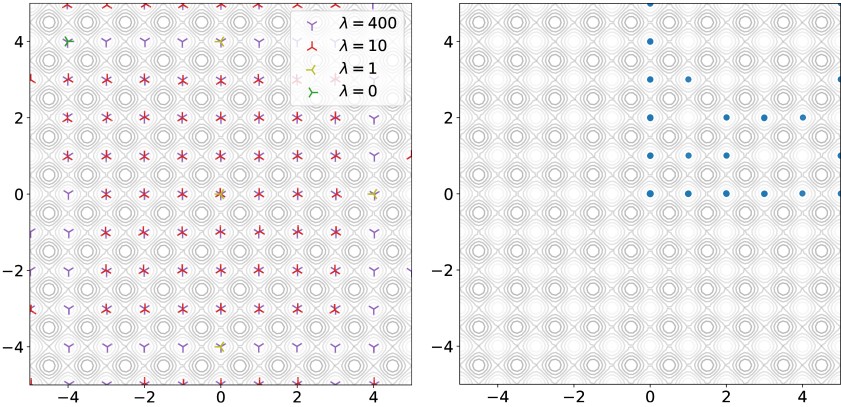

Figure D.2: Left: the result of POL after 10 iterations with $\lambda = 0, 1, 10, 400$ for the 2D cosine function which has weakly convex constant $\xi = 40\pi^2 < 400$. Right: particle swarms recovered by Cao et al. (2019) for after 10 iterations from 256 independent runs. The population size of the swarm is 4 (default value in their source code).

with $\xi = 40\pi^2 < 400$ and has global minima forming a grid (all local minima are global minima). On the left of Figure D.2, we see that when $\lambda = 400 > \xi$—in which case the condition of Theorem 3.1 is met—POL recovers all optima. In comparison, for $\lambda = 10$, the outer ring of solutions is missing, and with $\lambda = 0, 1$ most optima are missing in the grid.

To demonstrate how existing L2O methods can fail to recover multiple solutions, we conduct the same experiment on the L2O particle-swarm method by Cao et al. (2019), which recovers a swarm of particles that are close to optima. We use the default parameters in the provided source code except changing the objective to the 2D cosine function and the standard deviation of the initial random particles to 1. As the method by Cao et al. (2019) could produce particles outside $\mathcal{X} = [-5, 5]^2$, we add an additional term $0.01\|x\|^2$ to the objective $f(x)$; without such a term the particle swarm simply collapses to a single point far away from the origin. The results are shown on the right of Figure D.2. We see that even with 256 independent random starts and with population size 4, this method fails to recover most of the optima, in particular in non-positive quadrants.

## D.4 SAMPLING FROM CONIC SECTIONS

**Setup.** For this problem, the training dataset contains $2^{20}$ samples of $\tau \in \mathcal{T}$, while the test dataset has size 256. In our implementation and similarly in other benchmarks we do not store the dataset on disk, but instead generate them on the fly with fixed randomness. The $\tau$'s are sampled uniformly in $\mathcal{T}$. PD is run for $5 \times 10^4$ steps with learning rate 1.0. For step sizes, we choose $\lambda = 0.1$ for POL and $\eta = 1.0$ for GOL. We found that the training of GOL explodes when $\eta > 1.0$. Meanwhile, POL is able to take bigger ($1/\lambda = 10.0$) steps while staying stable during training (but might fail to recover

solutions due to large step size). To obtain solutions, we use 5 iterations for POL, while for GOL we use 100 iterations since it converges slower (and more iterations won't improve the results).

**Results.** We visualize for the conic section problem in Figure D.3 for 16 randomly chosen $\tau \in \mathcal{T}$. In Figure D.4 we plot of $\delta$ vs. $\text{WP}_t^\delta$ (5) to quantitatively verify how good POL and GOL are at recovering the level sets, where we treat the results by PD as the ground truth. Both visually and quantitatively, we see that POL outperforms GOL. Figure D.5 compares the convergence speed when applying the learned iterative operators at test time: clearly POL converges much faster.

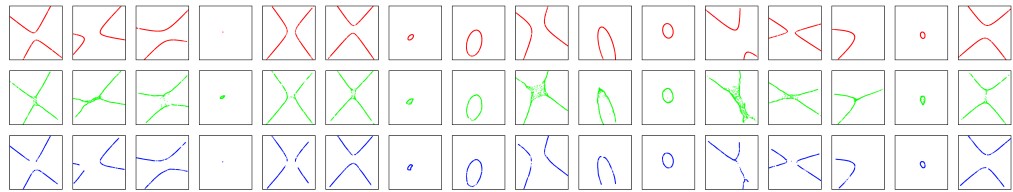

Figure D.3: Visualization of the solutions for the conic section problem. Red indicates the solutions by PD which we treat as ground truth. Green and blue indicate the solutions by GOL (Section 4) and POL (proposed method) respectively.

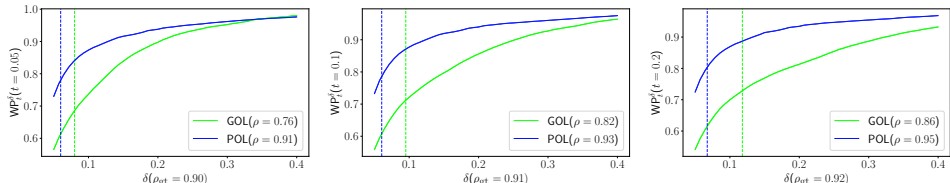

Figure D.4: The plot of $\delta$ vs. $\text{WP}_t^\delta$ for the conic section problem ($t = 0.05, 0.1, 0.2$). The vertical dashed line indicates $\text{WD}_t$. 50 equally spaced $\delta$ values are used to draw the plot. Here $\rho_{\text{gt}}$ indicates the percentage of PD solutions that have objectives $\leq t$, and $\rho$ similarly indicates the percentage of solutions for each method with objectives below $t$. We sample 1024 witnesses to compute $\text{WP}_t^\delta$, averaged over 256 test problem instances. The plot is averaged over 10 trials of witness sampling (the fill-in region's width indicates the standard deviation). Here the standard deviations are all less than $10^{-3}$ so the fill-in regions are too small to be visible.

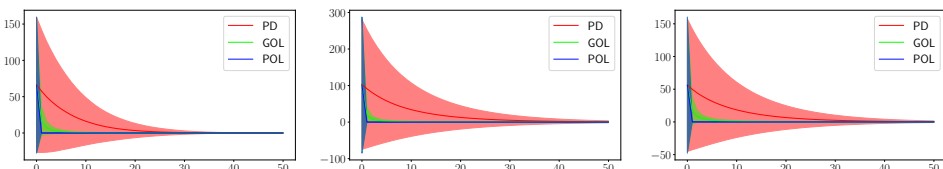

Figure D.5: Convergence speed comparison at test time for the conic section problem. For POL and GOL, the $x$-axis is the number of iterations used. For PD, the $x$-axis is the number of gradient descent steps, multiplied by 100. The horizontal axis shows the number of iterations, and the vertical axis shows the value of $f_\tau(x)$, averaged over all current solutions (fill-in region's width indicates standard deviation). The three plots shown correspond to the problem instances in the first three columns in Figure D.3. Once the operator has been trained, POL converges in less than 5 steps, while GOL converges slower (GOL is already trained with the largest step size without causing training to explode).

### D.5 NON-CONVEX SPARSE RECOVERY

**Setup.** For this problem, the training dataset contains 1024 samples of $\tau = (\alpha, p)$, while the test dataset has 128 samples. The $\tau$'s are sampled uniformly in $\mathcal{T} = [0, 1] \times [0.2, 0.5]$. We extract 4096 solutions from each method after training. For PD, we run $5 \times 10^5$ steps of gradient step with

learning rate $10^{-5}$. We found that due to the highly nonconvex landscape of the problem, bigger learning rates will cause PD to miss significant local minima. For step sizes, we choose $\lambda = 10$ for POL (so this corresponds to step size 0.1 for backward Euler) and $\eta = 0.1$ for GOL. To obtain solutions, POL requires less than 20 iterations to converge, while for GOL over 100 iterations are needed.

**Results.** We show the histogram of the solutions' objective values for PD, GOL, and POL in Figure D.7 for 4 problem instances. Figure D.6 visualizes the solutions for 8 problem instances projected onto the last two coordinates. GOL fails badly in all instances. Remarkably, despite the non-convexity of the problem and the much larger step size (0.1 compared to $10^{-5}$), POL yields solutions on par or better than PD when $p$ is small. For instance, for the second and third columns in Figure D.6 (corresponding to second and third columns in Figure D.7), PD (in red) misses near-optimal solutions that POL (in blue) captures. As such the results of PD can be suboptimal, so we do not compute witness metrics here.

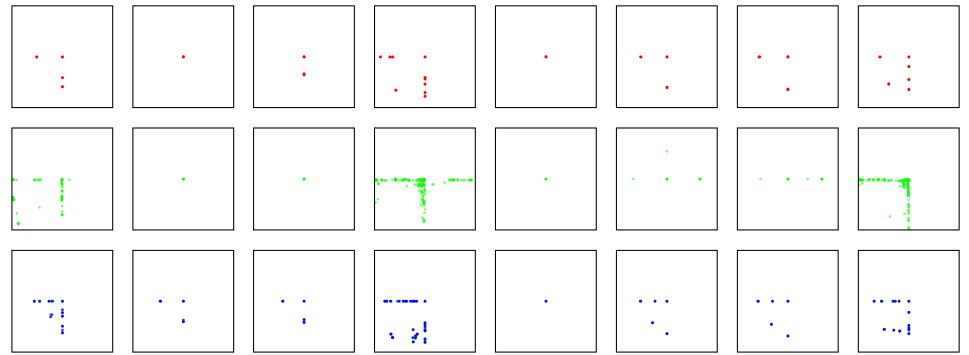

Figure D.6: Visualization of the solutions' objective values for the conic section problem. Red indicates the solutions by PD which we treat as ground truth. Green and blue indicate the solutions by GOL and POL (proposed method) respectively.

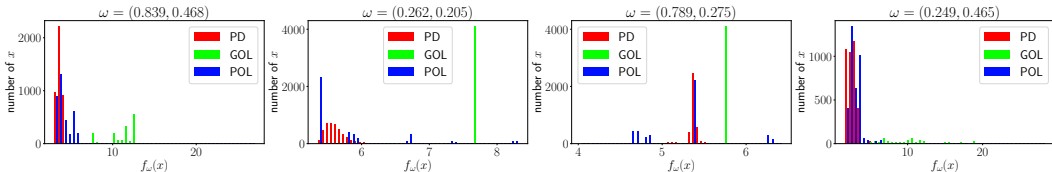

Figure D.7: Histograms of the objectives for non-convex sparse recovery on 4 problem instances. We denote $\tau = (\alpha, p)$ at the top, corresponding to a sparsity-inducing function of the form $\alpha \|x\|_p^p$.

**Comparison with proximal gradient descent.** For $p = 1/2, 2/3$, thresholding formulas exist for the $\ell^p$ norm (Cao et al., 2013). That is, the proximal operator of $\|x\|_p^p$ has a closed-form. This allows us to apply proximal gradient descent (Tibshirani et al., 2010) to solve (7). When $p = 1$ (i.e. when the problem reduces to LASSO), this reduces to the popular iterative soft-thresholding algorithm (ISTA) which converges significantly faster than gradient descent.

We compare the convergence speed of POL to that of proximal gradient descent (denoted PGD) for the $p = 1/2$ case. We also include PD for reference. The generation of data (i.e. $A$ and $y$ in (7)) is the same as before. For POL we use the same setup with $\lambda = 10$ as before (corresponding to step size 0.1) except we restrict $p$ to $1/2$ during training and we train only for 1000 steps (note the test-time $\alpha$ is unseen during training). For PGD we use 0.04 as the step size because using 0.05 for the step size would lead to divergence of PGD — the objective would go to infinity. For PD we use 0.05 as the step size. We run all three methods for 200 steps (for POL this corresponds to 200 steps of PPA after training) and visualize the convergence and histograms of the objective for each method in Figure D.8. We see that POL converges faster than PGD even when PGD is highly specialized to the $p = 1/2$ case (where the thresholding formula has a closed form).

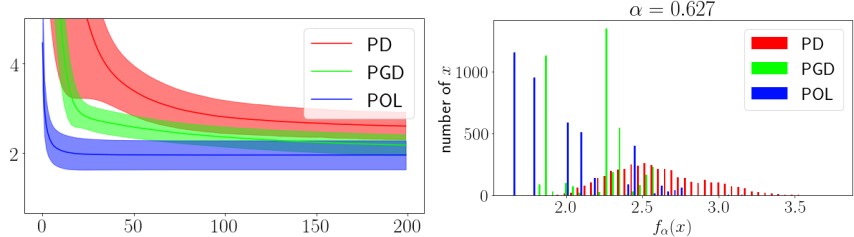

Figure D.8: Convergence (left figure) and histograms (right figure) of the objective for the $p = 1/2$ non-convex sparse recovery problem with $\alpha = 0.627$. For the convergence figure on the left, the horizontal axis shows the number of iterations, and the vertical axis shows the value of $f_\alpha(x)$, averaged over all current solutions (fill-in region's width indicates standard deviation). PGD denotes the proximal gradient descent method (Tibshirani et al., 2010) with the closed-form thresholding formula by Cao et al. (2013).

**Other sparsity-inducing regularizers.** Our method can be applied to sparse recovery problems with other sparsity-inducing regularizers in a straightforward manner. Consider minimax concave penalty (MCP) from Yang et al. (2020) defined component-wise as:

$$\text{MCP}(x; \tau) := \begin{cases} |x| - \tau x^2 & |x| \le \frac{1}{2\tau} \\ \frac{1}{4\tau} & |x| > \frac{1}{2\tau}, \end{cases}$$

which is $\tau$-weakly convex. We repeat the same setup as in Section 5.2 but with objectives

$$f_\tau(x) := \|Ax - y\|_2^2 + \sum_{i=1}^{d} \text{MCP}(x_i; \tau), \tag{13}$$

for $\tau \in [0.5, 2]$. Note PGD is viable to solve (13) because the proximal operator of MCP has a closed-form. We run PGD for $2 \times 10^4$ iterations with step size $10^{-4}$ to make sure it converges fully. We show the histogram of the solutions' objective values for PD, GOL, POL, and PGD in Figure D.9. Our results are consistent with those in Figure D.7: POL is on par with PD and significantly outperforms GOL. POL also performs better than PGD which is only applicable because the regularizer MCP has a closed-form proximal operator.

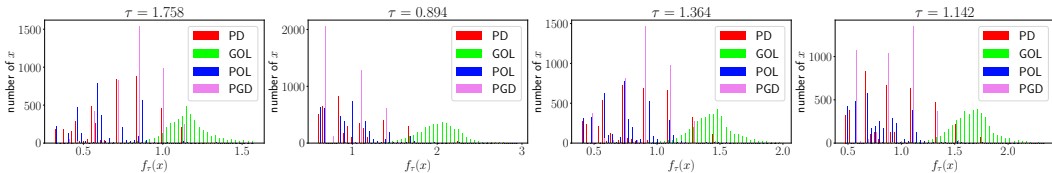

Figure D.9: Histograms of the objectives for sparse recovery with minimax concave penalty (MCP) on 4 problem instances.

## D.6 RANK-2 RELAXATION OF MAX-CUT

**Setup.** An additional feature of (8) is that the variables are constrained to $(S^1)^8 \subset \mathbf{R}^{16}$. Hence for POL and GOL we always project the output of the operator network to the constrained set (normalizing to unit length before computing the loss or before iterating), while for PD we apply projection after each gradient step.

We generate a training dataset of $2^{20}$ graphs and a test dataset of 1024 graphs using the procedure described in Section 5.3: half of the graphs will be Erdős-Rényi graphs with $p = 0.5$ and the remaining half being $K_8$ with edge weights drawn from $[0, 1]$ uniformly. For PD, we use learning rate $10^{-4}$. For step sizes of POL and GOL, we choose $\lambda = 10.0$ and $\eta = 10.0$.

We choose to directly feed the edge weight vector $\tau \in \mathbf{R}^{28}$ to the operator network (Appendix B). We find this simple encoding works better than alternatives such as graph convolutional networks.

This is likely because $x \in \mathcal{X} = (S^1)^8$ requires order information from the encoded $\tau$, so graph pooling operation can be detrimental for the operator network architecture. Designing an equivariant operator network that is capable of effectively consuming larger graphs is an interesting direction for future work.

**Results.** If a cut happens to be a local minimum of the relaxation, then it is a maximum cut (Theorem 3.4 of Burer et al. (2002)). However, finding all the local minima of the relaxation is not enough to find all max cuts as max cuts can also appear as saddle points (see the discussion after Theorem 3.4 of Burer et al. (2002)). Hence solving the MSO (8) is not enough to identify all the max cuts. Nevertheless, we can still compare POL and GOL against PD based solely on the relaxed MSO problem corresponding to the objective (8). In Figure D.10, we plot $\delta$ vs. $\mathrm{WP}_t^\delta$ (5) to verify the quality of

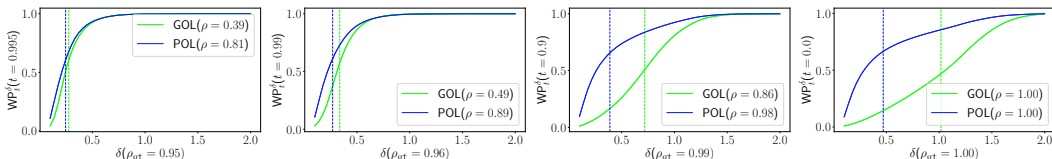

Figure D.10: The plot of $\delta$ vs. $\mathrm{WP}_t^\delta$ for the max-cut problem. See the caption of Figure D.4 for the meaning of the symbols. As different edge weights lead to different minima values, we choose the threshold $t$ in a relative manner: the actual threshold used will be $t$ times the best objective value found by PD.

the solutions obtained by POL and GOL compared to PD. We see that POL more faithfully recovers the solutions generated by PD with consistently higher witnessed precision.

Empirically, we found the proposed POL can identify a diverse family of cuts. We visualize the multiple cuts obtained by POL for a number of graphs in Figure D.12. Although some cuts are not maximal, they are likely due to the relaxation — not all fractional solutions correspond to a cut — and not because of the proposed method. As evident in Figure D.10, they are still very close to the local minima of (8) generated by PD.

### D.7   SYMMETRY DETECTION OF 3D SHAPES

**Setup.** Since the variables in (9) is constrained to $\mathcal{X} = S^2 \times \mathbf{R}_{\geq 0}$, we always project the output of the operator network to the constrained set: for $x = (n, d) \in \mathcal{X}$, we normalize $n$ to have unit length and take absolute value of $d$. The same projection is applied after each gradient step in PD.

To generate training and test datasets, we use the original train/test split of the MCB dataset (Kim et al., 2020) but filter out meshes with more than 5000 triangles and keep up to 100 meshes per category to make the categories more balanced. During each training iteration, a fresh batch of point clouds are sampled (these are $\tau$'s) from the meshes in the current batch. For step sizes, we choose $\lambda = 1.0$ for POL and $\eta = 10.0$ for GOL. The training of POL and GOL takes about 30 hours. For PD, we run gradient descent for 500 iterations for each model, which is sufficient for convergence.

We use the official implementation of DGCNN by Wang et al. (2019) as the encoder with the modification that we change the input channels to 6-dimension to consume oriented point clouds and we turn off the dropout layers which do not improve performance.

The objective (9) involves $s_\tau$ which requires point-to-mesh projection. We implemented custom CUDA functions to speed up the projection. Even so, it remains the bottleneck of training. Since GOL requires multiple evaluations, it is extremely slow and can take more than a week. As such, we set $Q = 1$ in (6). Both POL and GOL are trained for $10^5$ iterations with batch size 8. At test time iterating the operator networks does not need to evaluate the objective nor the $s_\tau$'s; moreover, only point clouds are needed.

**Results.** We show the witness metrics in Figure D.11; quantitatively, POL exhibits far higher witnessed precision values than GOL. We show a visualization of iterations of PPA with the learned proximal operator in Figure D.13. In particular, our method is capable of detecting complicated discrete reflectional symmetries as well as a continuous family of reflectional symmetries for cylindrical objects.

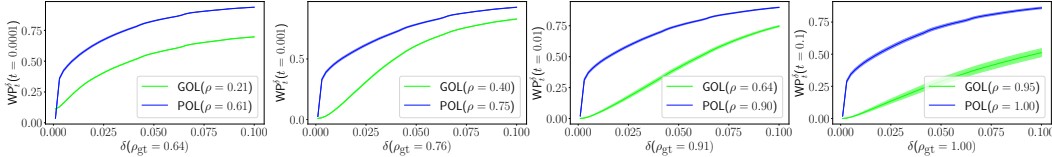

Figure D.11:   The plot of $\delta$ vs. $\mathrm{WP}_t^\delta$ for symmetry detection on the test dataset of Kim et al. (2020) ($t = 10^{-4}, 10^{-3}, 10^{-2}, 10^{-1}$). See the caption of Figure D.4 for the meaning of the various notations. We do not show $\mathrm{WD}_t$ as the vertical bars here because GOL's $\mathrm{WD}_t$ is much higher than POL's and is out of the range for the horizontal axis. We sample 1024 witnesses to compute $\mathrm{WP}_t^\delta$, averaged over 10 trials of witness sampling (the fill-in region's width indicates the standard deviation).

## D.8   OBJECT DETECTION IN IMAGES

**Setup.**  We use the training and validation split of COCO2017 (Lin et al., 2014) as the training and test dataset, keeping only images with at most 10 ground truth bounding boxes. For training, we use common augmentation techniques such as random resize/crop, horizontal flip, and random RGB shift, to generate a $400 \times 400$ patch from each training batch image, with batch size 32. For evaluation, we crop a $400 \times 400$ image patch from each test image. For step sizes, we choose $\lambda = 1.0$ for POL and $\eta = 1.0$ for GOL. We train both POL and GOL for $10^6$ steps. This takes about 100 hours. To extract solutions, we use 100 iterations for POL (for most images it only needs 5 iterations to converge) and 1000 iterations for GOL (the convergence is very slow so we run it for a large number of iterations).

We fine-tune PyTorch's pretrained ResNet-50 (He et al., 2016) with the following modifications. We first delete the last fully-connected layer. Then we add an additional linear layer to turn the 2048 channels into 256. We then add sinusoidal positional encodings to pixels in the feature image output by ResNet-50 followed by a fully-connected layers with hidden layer sizes $256, 256, 256$. Finally average pooling is used to obtain a single feature vector for the image.

For Faster R-CNN (FRCNN), we use the pretrained model from PyTorch with ResNet-50 backbone and a regional proposal network. It should be noted that FRCNN is designed for a different task that includes prediction of class labels, and thus it is trained with more supervision (object class labels) than our method and it uses additional loss terms for class labels.

For the alternative method FN that predicts a fixed number of boxes, we attach a fully-connected layer of hidden sizes $[256, 256, 80]$ with ReLU activation to consume the pooled feature vector from ResNet-50. The output vector of dimension 80 is then reshaped to $20 \times 4$, representing the box parameters of 20 boxes. We use chamfer distance between the set of predicted boxes and the set of ground truth boxes as the training loss.

**Results.**  In Table 1, we compute witness metrics and traditional metrics including precision and recall. As our method does not output confidence scores, we cannot use common evaluation metrics such as average precision. To calculate precision and recall, which normally would require an order given by the confidence scores, we instead build a bipartite graph between the predicted boxes and the ground truth, adding an edge if the Intersection over Union (IoU) between two boxes is greater than $0.5$. Then we consider predictions that appear in the Hungarian max matching as true positives, and the unmatched ones false positives. Then precision is defined as the number of true positives over the total number of predictions, while recall is defined as the number of true positives over the total number of ground truth boxes. When computing metrics for POL and GOL, we run mean-shift algorithm with RBF bandwidth $0.01$ to find the centers of clusters and use them as the predictions. As shown in Figure 5, the clusters formed by POL are usually extremely sharp after a few steps, and any reasonable bandwidth will result in the same clusters.

In Figure D.14, we show the detection results by our method for a large number of test images chosen at random.

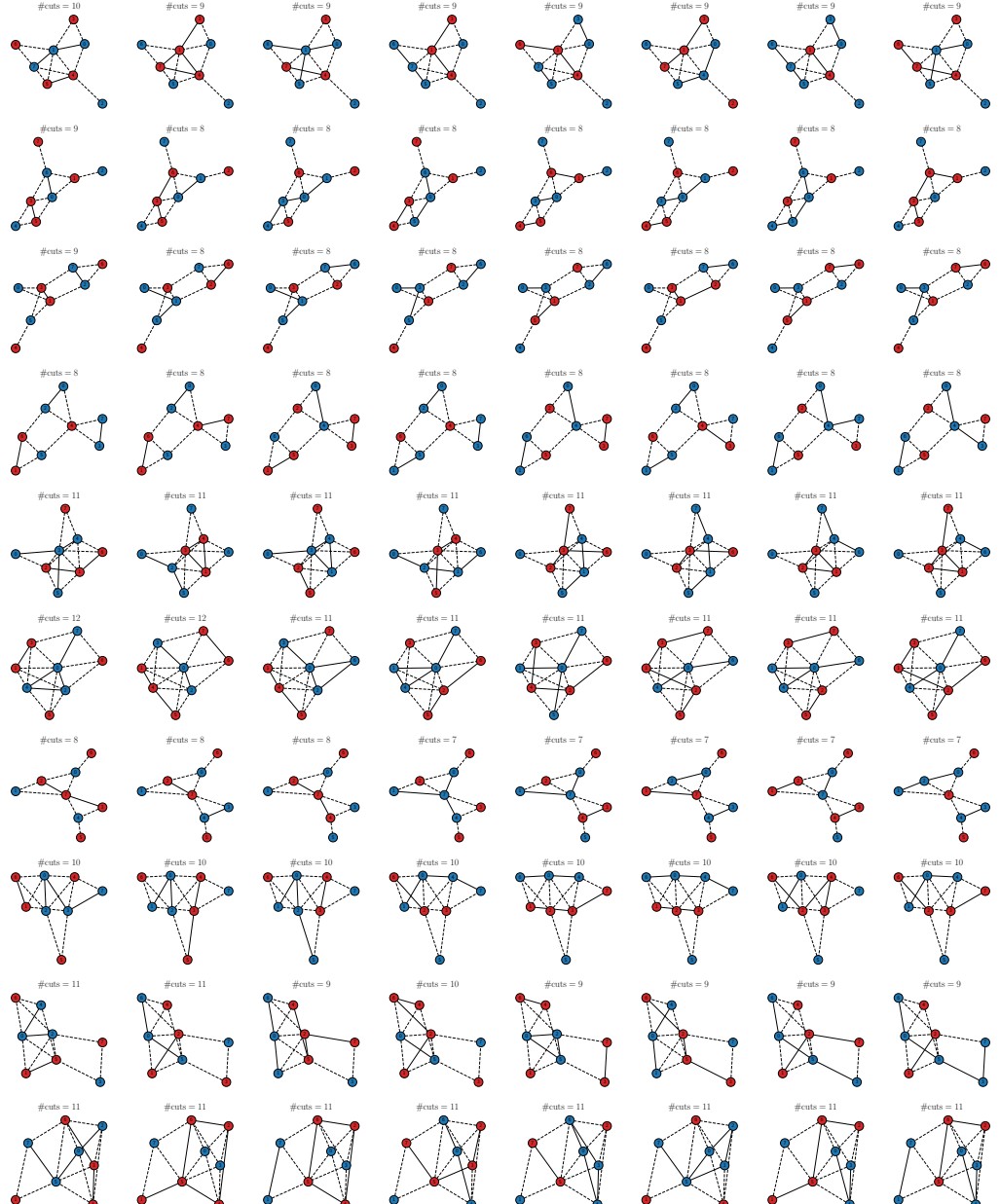

Figure D.12: Visualization of the cuts obtained by applying Goemans-Williamson-type procedure to randomly selected solutions of POL. The graphs are chosen among the ones that have at least 8 solutions uniformly at random without human intervention. Each row contains the multiple solutions for the same graph with binary weights. Two colors indicate the two vertex sets separated by the cut. Dashed lines indicate edges in the cut. For each graph we annotate the number of cuts on top. The 0th vertex is always in blue to remove the obvious duplicates obtained by swapping the two colors on each vertex.

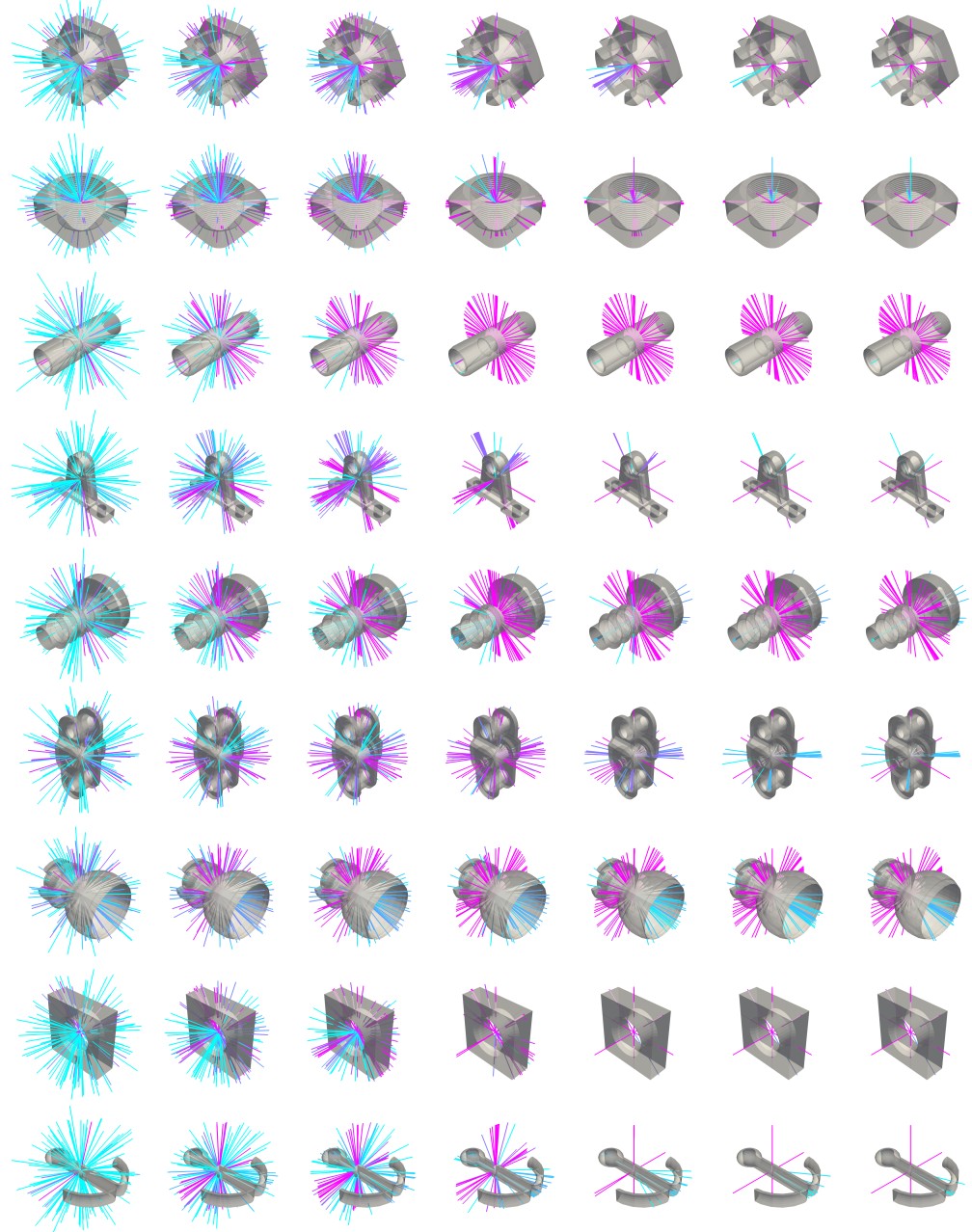

Figure D.13: Visualization of PPA with learned proximal operator on selected models from the test dataset of Kim et al. (2020). Iterations 0, 1, 2, 5, 10, 15, 20 are shown, where the 0th iteration contains the initial samples from unif($\mathcal{X}$). Pink indicates lower objective value in (9), while light blue indicates higher.

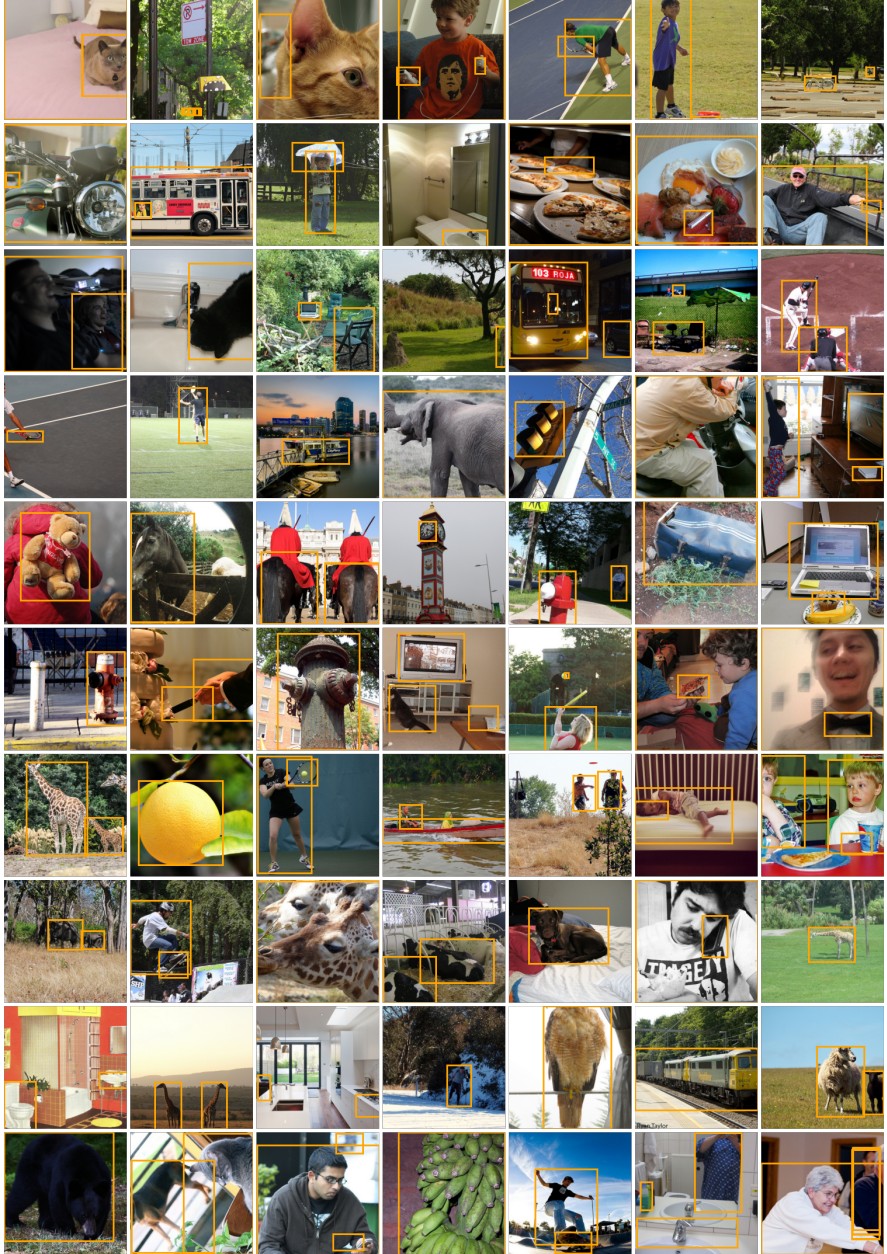

Figure D.14: Randomly selected object detection results by POL on COCO17 validation split. Each test image is $400 \times 400$ patch cropped from the original image with a random scaling by a number between $0.5$ and $1$. In each image we display all $1024$ bounding boxes without clustering, most of which are perfectly overlapping. For some images there is a bounding box for the whole image (check if the image has an orange border). There is no class label associated with each box.

# E  CONNECTION TO WASSERSTEIN GRADIENT FLOWS

In this section we show that we can view (2) as solving the JKO discretization of Wasserstein gradient flows at every time step, under the assumption that the measures along the JKO discretization are absolutely continuous.

If $\mathcal{F}(\mu)$ is a linear functional of the form $\mathcal{F}(\mu) = \int f \mathrm{d}\mu$ on $\mathcal{P}(\mathcal{X})$, the space of probability distributions in $\mathcal{X}$ with $\mathcal{X}$ compact, then the JKO discretization of the gradient flow of $\mathcal{F}$ at step $t+1$ with step size $1/\lambda$ is

$$\mu_{t+1} = \underset{\mu \in \mathcal{P}_2(\mathcal{X})}{\arg\min} \left\{ \int f \mathrm{d}\mu + \frac{\lambda}{2} W_2^2(\mu_t, \mu) \right\},$$

where $W_2$ is the Wasserstein-2 distance and we assume $\mu_t$ is absolutely continuous. Let

$$\mathcal{F}(\mu) := \int f \mathrm{d}\mu + \frac{\lambda}{2} W_2^2(\mu_t, \mu).$$

Let us also define another functional such that for a Borel map $T : \mathcal{X} \to \mathcal{X}$,

$$\mathcal{G}(T) := \int f(T(x)) \mathrm{d}\mu_t(x) + \frac{\lambda}{2} \int \|T(x) - x\|_2^2 \mathrm{d}\mu_t(x).$$

First given $\mu \in \mathcal{P}(\mathcal{X})$, since $\mathcal{X}$ is compact (so in particular all probability distributions have finite second moments), by Brenier's theorem (Ambrosio et al., 2005, Theorem 6.2.4), there exists a Borel map $T$ (the *Monge map*) such that $T_{\#}\mu_t = \mu$ and $W_2^2(\mu_t, \mu) = \int \|T(x) - x\|_2^2 \mathrm{d}\mu_t(x)$. Hence for such $\mu$ and $T$ we have $\mathcal{G}(T) = \mathcal{F}(\mu)$, and thus $\min_{\mu \in \mathcal{P}_2(\mathbf{R}^d)} \mathcal{F}(\mu) \geq \min_T \mathcal{G}(T)$.

Next given a Borel $T$, let $\mu = T_{\#}\mu_t$. By Brenier's theorem, let $T'$ be the Monge map corresponding to $W_2(\mu_t, \mu)$ so that $\mu = T'_{\#}\mu_t$ and $\int \|T'(x) - x\|_2^2 \mathrm{d}\mu_t(x) = W_2(\mu_t, \mu) \leq \int \|T(x) - x\|_2^2 \mathrm{d}\mu_t(x)$. This shows that $\mathcal{F}(\mu) \leq \mathcal{G}(T)$ and hence $\min_{\mu \in \mathcal{P}_2(\mathbf{R}^d)} \mathcal{F}(\mu) \leq \min_T \mathcal{G}(T)$. Thus $\min_{\mu \in \mathcal{P}_2(\mathbf{R}^d)} \mathcal{F}(\mu) = \min_T \mathcal{G}(T)$.

If $\mu_t$ has full support, then the best $T^* := \arg\min_T \mathcal{G}(T)$ is obtained pointwise and it becomes the proximal operator of $f$ (cf. (2)). In particular, $T^*$ does not depend on $\mu_t$.

