# OpenReview forum: "Learning Proximal Operators to Discover Multiple Optima"
_ICLR.cc/2023/Conference — ICLR 2023 poster_

### Official Review · Reviewer_T3qd · 2022-10-25

**Confidence:** 3
**Correctness:** 4
**Technical Novelty And Significance:** 3
**Empirical Novelty And Significance:** 3
**Recommendation:** 8

**Clarity, Quality, Novelty And Reproducibility:**

The paper is written clearly and ideas are novel. I do not have many comments, I think this is a good contribution in general enabling further research.

1) In my opinion, there could be more experiments regarding recovering true proximal operator in different (more challenging settings than $\ell_1$) and especially to demonstrate whether there is any issue with increasing dimension of the problem.

2) Sampling from $\text{Unif}(\mathcal{X})$ does not look like the most efficient way to train the objective proposed in (2) and can certainly be very difficult. Can authors give their opinion about this or whether it makes sense to use another measure or technique to estimate the loss?

3) As the problem gets higher dimensional, how is the number of samples needed to approximate the loss affected?

**Strength And Weaknesses:**

Strength: The formulation is neat and clean -- the paper is well written. On top of that, theoretical results are provided which is rare in learning to optimise literature.

Weakness: Would be great if more experiments regarding approximation of true proximal operators were done.

**Summary Of The Paper:**

This paper proposes a method to learn proximal mappings which have multiple beneficial properties as optimisers, alternative to gradient descent and variants.

**Summary Of The Review:**

The paper reads well and is motivated well. Proximal operators tend to perform much better and more stable than gradient updates, therefore learning prox operators is a viable idea for improving "learning to optimise" tasks.

---

> ### Author Response · Authors · 2022-11-08
> **Author's response**
>
> Thank you for your positive review of our work. We are glad that you like our formulation and the theoretical results.
>
> We point out that as written in the introduction and acknowledged in the conclusion, the current paper assumes $x$ is in a low-dimensional space. However, we allow $\tau$ to be in a very high-dimensional space. This already allows for applications like symmetry detection (Section 5.4) and object detection (Section 5.5). In our response below, we detail some potential solutions to scale up our method to higher dimensions, but we wish to underscore the practicality and value of our method even in the current setting.
>
> **In my opinion, there could be more experiments regarding recovering true proximal operator in different (more challenging settings than $\ell_1$) and especially to demonstrate whether there is any issue with increasing dimension of the problem.**
>
> In Proposition A.2 (second half of Theorem 3.1), we show that the mean-squared error of the learned proximal operator is linear in the training loss. While it would be exciting to include  more experiments testing whether the true proximal operator is recovered, there are not many functions (especially non-convex ones) for which we know the closed-form proximal operators. If we use another algorithm to approximate the true proximal operator, we will need to account for the approximation error introduced there.
>
> **Sampling from $\text{Unif}(\mathcal{X})$ does not look like the most efficient way to train the objective proposed in (2) and can certainly be very difficult. Can authors give their opinion about this or whether it makes sense to use another measure or technique to estimate the loss?**
>
> This is a very interesting question. The choice of $\text{Unif}(\mathcal{X})$ is only viable because we assume $\mathcal{X}$ is of low dimension. We believe our formulation (2) can be extended to handle higher dimensional $\mathcal{X}$ (as a low-dimensional manifold embedded in high-dimensional space) for which we have data access.
>
> ***Here we outline a potential extension of our method to image optimization problems***. Suppose $\mathcal{X}$ is the space of images and we have a dataset $\mathcal{D}$ of training images. Then, we can sample from $\mathcal{D}$ instead of from a uniform distribution when optimizing (2). In addition, we need to augment $\Phi$ so that its output lies on the image manifold, so we can replace $\Phi$ by $\pi \circ \Phi$ where $\pi$ is a projection operator (e.g. denoiser from diffusion models). For instance, we can take $\tau$ to be a text caption and $f_\tau$ to be the CLIP (Contrastive Language–Image Pre-training) score. Then the learned $\Phi$ is capable of generating multiple images that conform to a given caption.
>
> To sum up, we believe it is possible to scale up our method if we can sample from $\mathcal{X}$ (e.g. with data access) and can project back onto $\mathcal{X}$ efficiently. We save this interesting extension of our algorithm for future work.
>
> **As the problem gets higher dimensional, how is the number of samples needed to approximate the loss affected?**
>
> If using uniform sampling over $\mathcal{X}$ like we did, we might need to use a bigger batch size to reduce variance when evaluating (2) during training. We have experienced that using too small of a batch size can result in longer training time. As mentioned above, studying the proposed method and its extension to higher dimensions will be a potentially fruitful future direction.

---

> ### Author Response · Authors · 2022-11-14
> **Could you please check our response?**
>
> Dear Reviewer,
>
> Since only a few days remain in the discussion period, we would appreciate it if you check and reply to our response to your comments soon. This will give us time to address further questions and comments that you may have before the end of the discussion period. If our response adequately addresses your concerns, please consider raising the score of our submission. Thank you very much for your time.

---

### Official Review · Reviewer_4Grh · 2022-10-25

**Confidence:** 3
**Correctness:** 4
**Technical Novelty And Significance:** 3
**Empirical Novelty And Significance:** 3
**Recommendation:** 8

**Clarity, Quality, Novelty And Reproducibility:**

The paper is written very clearly, with unambiguous notation and ample detail
(both mathematical and engineering-type). The contribution of a general
framework and a few baselines for general MSO problems seems to be novel.



**Strength And Weaknesses:**

## Strengths

The authors present a method applicable to general MSO problems that strikes a
good balanced between computational/optimization principles (beginning with the
proximal point iteration) and learning-based approaches (learning a proximal
operator on datasets). This suggests the ability to provide theoretical
guarantees for the approach.

The experiments are extensive, and strike a good balance between synthetic
tasks (which allow specific behaviors of the method to be understood) and
important applications (symmetry detection and object detection on big
datasets, comparing to reasonable benchmarks). Ablations and comparisons to
other general MSO baselines are given in the main text and the extensive
experimental appendices. It seems reasonable to imagine that with further
tuning, the approach could be even more competitive relative to strong
baselines.


## Weaknesses

References to a large body of literature on learning proximal operators for
inverse problems (esp. in imaging) seem to be omitted. See e.g. [1-3] below --
and note that in particular the paradigm in many of the most recent works of
this line is different from that of LISTA (which is referenced and discussed),
where the optimization problem is used to derive an architecture. These
comparisons would be helpful in situating the present work in this broader
context (there may be intuitions developed here that can be applied in those
settings, and vice versa).

[1] http://arxiv.org/abs/2102.07944

[2] http://dx.doi.org/10.1109/JSAIT.2020.2991563

[3] https://arxiv.org/abs/2209.04504

The assumption of a Lipschitz gradient in Theorem 3.1 does not seem to make
sense to me in the context of multi-solution optimization. For instance,
considering the example of bounding box detection given by the authors in the
introduction where $f_{\tau}(x)$ is the minimum distance of the bounding box
$x$ to a ground truth box in the image $\tau$, at any time where there are
multiple boxes in the image $\tau$, there will be boxes $x'$ that are at an
equal distance from two or more ground truth boxes, implying that $x'$ is a
point of discontinuity for $\nabla f_{\tau}$ (and therefore the gradient is not
Lipschitz). More generally, it seems like this issue should arise any time
$f_{\tau}$ is a metric and there are multiple minima (this feels like a result
from geometry, involving regularity of the distance function at the cut locus).
The authors may want to apply a global convergence theorem that does not
require smooth activation functions and Lipschitz gradients, as Kawaguchi and
Huang's does -- for example, Du et al., or Allen-Zhu et al, which are
formulated for networks with ReLUs (hence non-Lipschitz gradients are an
inherent part of the problem). Some discussion of this assumption in section
3.3 seems warranted (I did not see it discussed anywhere -- the authors discuss
GD vs. SGD, the network architecture, weak convexity).

For the experiments on symmetry detection, it would be
interesting to see how the approach compares to strong baselines that are
specifically tuned for these tasks (rather than just POL vs. GOL). For object
detection, the comparisons to Faster R-CNN are very interesting and seem
promising, but would comparisons to more recent networks like YOLO (etc.) be
possible also?

## Questions / Minor Points

I am curious whether other baseline approaches were attempted than just the one
proposed at the bottom of page 5 -- it seems to me like this proposed approach
might be more challenging to optimize than the proximal loss (2), which is like
approximating a single map which is iterated at test time, given that here we
need to approximate $Q>1$ iterations of the gradient map in general (which
seems like it might depend a lot on the architecture of the network $\Psi$
being suitable). Is it possible to solve this problem for $Q=1$, then iterate
the learned mapping to simulate many steps of gradient descent (possibly with
some stability-promoting regularization?), or the like?
I did not understand why the "particle descent" approach on the previous page
is "not comparable" -- can't one just run gradient descent from many random
initializations given a "test" $\tau$?

For the sparse recovery experiment, it could be that learning in this setting
has an advantage over GD due to the low-dimensionality ($m=4$, $d=8$) of the
experimental setup.

Why are DGCNN and ResNet used in the symmetry detection and object detection
experiments as encoders for the raw data? Is there a restriction that not only
$x$ but also $\tau$ should be low-dimensional here?


**Summary Of The Paper:**

The authors consider the general problem of multi-solution optimization (MSO),
where one is interested in obtaining a sampling (perhaps all) of the minimizers
of each of a family of (nonconvex) functions $\{ f_{\tau} : \tau \in \mathcal{T} \}$
defined on a low-dimensional domain $\mathcal{X}$ (here, $\mathcal{T}$
represents parameters -- an example given is bounding box detection, where
$\mathcal{T}$ is a class of digital images and $\mathcal{X}$ is the
four-dimensional euclidean space of bounding box parameters). The authors
propose a method for this problem that consists of learning a proximal operator
(parameterized by a neural network) for the functions $f_{\tau}$
(assuming access to ground truth) over a training set of parameters $\tau$ and
inputs $x$ (the loss/training setup is similar to what one would formulate for
multi-task learning); at test time (given a new $\tau$), a set of putative
solutions is generated by repeatedly iterating the learned operator from
various random initializations (in an attempt to simulate the proximal point
algorithm). The authors propose an evaluation metric for these families of
putative solutions in general MSO problems, given ground truth (it is described
as a weighted chamfer distance between the putative set and the ground truth
set, with weights proportional to a certain Voronoi tiling of the space
$\mathcal{X}$ -- hence more robust to outliers than standard metrics). They
compare their approach to a benchmark of gradient descent from repeated random
initialization and an approach that attempts to learn an operator that
approximates a finite number of gradient descent iterations (trained similarly
to theirs) on several synthetic and practical MSO tasks, showing that their
approach performs well.



**Summary Of The Review:**

The method is clearly presented, and the paper seems to do a useful service
of organizing lots of thought and baseline approaches to general MSO problems.
The approach taken here to learning a proximal operator and then applying it as
in the proximal point iteration has been studied elsewhere, but this study in
the MSO context seems new and useful. The theory presented here seems mostly
"plug and play" with regards to existing results, so it could be changed to a
MSO-relevant context by applying a different result from the literature.

---

> ### Author Response · Authors · 2022-11-08
> **Author's Response (1/2)**
>
> Thank you for your thoughtful review. We greatly appreciate your recognition of the balance between optimization principles and learning-based approaches and of our extensive set of experiments. Please find below our responses to your questions and comments.
>
> **References to a large body of literature on learning proximal operators for inverse problems (esp. in imaging) seem to be omitted**
>
> Thank you for the additional references. We have incorporated them in the related works section.
>
> We emphasize that deep unrolling models and deep equilibrium models require ground truth supervision (i.e. clean images). The goal of training in these methods is to learn the parameters of the neural networks (e.g. in the regularizer) involved in a fixed unrolling algorithm. For instance, in Eq (13) of [1] they use an MSE loss that requires access to training measurement/image pairs. In contrast, our method requires only access to the objective function; we do not need ground truth for the inverse problems studied in Section 5.2.
>
> To put our paper in the taxonomy of [2], our method belongs to their 4.1.2 (only having access to measurements during training), which describes a few denoising methods using neural networks. A possible direction for future work might explore whether we can apply our techniques to denoise images, for example as an alternative to popular diffusion models. The current paper, however, develops a generic framework for solving MSO, and solving inverse problems is only one application.
>
> **The assumption of a Lipschitz gradient in Theorem 3.1 does not seem to make sense to me in the context of multi-solution optimization**
>
> Thank you for pointing this out. Below Theorem 3.1 of the revision, we added acknowledgment that the Lipschitz-gradient condition is not met in some applications.
>
> We chose Kawaguchi and Huang because the amount of parameterization needed in their result is only linear in the amount of training data, unlike other works. Regarding more general convergence results, we could not find results in the literature that allow non-Lipschitz gradients. [Allen-Zhu et al 2019] require the objective to be Lipschitz smooth (see the conditions before their Theorem 6) even though they allow ReLU activation. [Du et al. 2018] are only concerned with a quadratic loss. *We would greatly appreciate if the reviewer could suggest a more relevant reference.*
>
> **For the experiments on symmetry detection, it would be interesting to see how the approach compares to strong baselines that are specifically tuned for these tasks (rather than just POL vs. GOL)**
>
> To the best of our knowledge, there is no existing learning-based method nor benchmark for unsupervised symmetry detection that allows an arbitrary number of planar symmetries. In the two papers that we cited, [Shi et al. 2020] require supervision of ground truth symmetries that are obtained by human annotators. The method by [Gao et al. 2020] is unsupervised but can only detect a few symmetries (all examples in their paper have at most 3 planar symmetries), whereas in our Figure 4 our method can detect many-fold symmetries. We have clarified this point in Section 5.4 of the revision.
>
> **For object detection, the comparisons to Faster R-CNN are very interesting and seem promising, but would comparisons to more recent networks like YOLO (etc.) be possible also?**
>
> We are aware that there are more advanced object-detection methods like newer versions of YOLO. A central limitation of YOLO is that it predicts many duplicates (YOLO first divides the image into grids, and then for each grid, it predicts many boxes), and non-maximal suppression is needed to get rid of most boxes. In contrast, our method does not need such postprocessing, as evident in Figure 5 and Figure D.13 (no clustering is done).
>
> Granted our numbers are lower than SOTA for object detection, but our method is extremely simple and clean, simply using a ResNet-50 encoder (that takes the entire image) with a general-purpose operator network without any special treatment of regional proposals (FasterRCNN) or subdivision (YOLO). There is substantial room for improvement to combine our method with insights from SOTA object detection methods, which we save for more specialized future work.

---

> ### Author Response · Authors · 2022-11-08
> **Author's Response (2/2)**
>
> **Is it possible to solve this problem for $Q=1$, then iterate the learned mapping to simulate many steps of gradient descent (possibly with some stability-promoting regularization?), or the like?**
>
> Indeed GOL is more challenging to optimize than POL with the proximal loss (2) — this again shows the effectiveness of our formulation.
>
> As mentioned in the first paragraph of Section D.4, using too big a step size can make GOL fail to train, whereas empirically POL is much more robust. We have tried with $Q=1$ and found we need to iterate the learned operator too many times, which can result in instability. For instance, in Section D.4, even with $Q=10$, we will need 100 iterations for GOL, whereas for POL we only need 5 iterations to obtain good solutions.
>
> We are not sure what kind of “stability-promoting regularization” can be used to improve GOL that will not incur overdamping (and hence result in incorrect solutions); we would appreciate it if the reviewer can share suggestions for coping with this issue.
>
> **I did not understand why the "particle descent" approach on the previous page is "not comparable" -- can't one just run gradient descent from many random initializations given a "test" $\tau$**
>
> At test time, we have access to only $\tau$ but not $f_\tau$, so particle descent cannot be used for a fair comparison.
>
> In a sense, however, we are comparing to particle descent (treated as ground truth, but to obtain the ground truth it needs access to $f_\tau$) because the witness metrics are all computed between the particle descent results and the results by GOL/POL.
>
> **For the sparse recovery experiment, it could be that learning in this setting has an advantage over GD due to the low-dimensionality of the experimental setup.**
>
> The applications in 5.1, 5.2, and 5.3 are intended as lower-dimensional, controllable examples. We keep the dimensions small so that we can approximate ground truth using particle descent reliably.
>
> As mentioned in our conclusion, we also assume $x$ lives in a low-dimensional space. It is an interesting future direction to see how our method performs in higher dimensions such as in images. In our response to reviewer T3qd, we outlined an exciting extension of our method to optimization problems in images.
>
> **Why are DGCNN and ResNet used in the symmetry detection and object detection experiments as encoders for the raw data? Is there a restriction that not only $x$ but also $\tau$ should be low-dimensional here?**
>
> We choose DGCNN and ResNet for encoding because they are standard architectures for point clouds and images, respectively. *In both cases, $\tau$ is very high-dimensional*: for point clouds, the dimension of $\tau$ is $3\times N$ where $N$ is the number of points, and for images, it is $3 \times W \times H$ for width $W$ and height $H$. Our experiments show that our method is effective if $\tau$ is high-dimensional.

---

> ### Author Response · Authors · 2022-11-14
> **Could you please check our response?**
>
> Dear Reviewer,
>
> Since only a few days remain in the discussion period, we would appreciate it if you check and reply to our response to your comments soon. This will give us time to address further questions and comments that you may have before the end of the discussion period. If our response adequately addresses your concerns, please consider raising the score of our submission. Thank you very much for your time.

---

> > ### Comment · Reviewer_4Grh · 2022-12-06
> > **thanks**
> >
> > Dear authors,
> >
> > Thanks for your detailed response. Let me respond to a few points below:
> > - [related works on unrolling] Thanks for this clarification, this speaks well to your work's contribution.
> > - [neural net global convergence] My impression is that Allen-Zhu's Theorem 6 is suitable for your application, but please correct me if I'm wrong -- I think their Theorem 6 only requires the loss to be Lipschitz as a function of the network's output, and shouldn't this be reasonable for natural MSO objectives? (The bounding box example seems okay here, and more generally objectives which are e.g. defined as a distance function to some landmarks on a manifold should be 1-Lipschitz by general considerations too.) Their Theorem 6 seems to also claim in a side remark that it applies to general gradient-bounded losses too (i.e. dropping the Lipschitz assumption), but it seems to me that it might require a careful chase through the proofs to verify this is indeed correct. I thank the authors for complaining about my complaint here, as checking the recent literature it does seem to be an unresolved issue to understand global convergence for general losses with the 'right' complexity: I was thinking of the recent works https://proceedings.neurips.cc/paper/2021/file/5d9e4a04afb9f3608ccc76c1ffa7573e-Paper.pdf and https://arxiv.org/pdf/2209.15106.pdf here, but both of these require smooth activations! At worst (i.e. if Allen-Zhu et al's result is really insufficient for MSO objectives), I think the discussion the authors have added to the revision will thus suffice completely in this connection.
> > - [Experimental aspects] Thanks for the clarifications here. Re: the GOL question, I agree with you that the natural regularization schemes I might have had in mind would lead to "overdamping".
> > I am satisfied with the clarifications and the quality of the work and will update my score.

---

> > > ### Author Response · Authors · 2022-12-08
> > > **Thank you**
> > >
> > > Dear reviewer,
> > >
> > > Thank you very much for acknowledging our work's contribution and updating the score!
> > >
> > > Re: Allen-Zhu's Theorem 6, we think they require the objective functions to be Lipschitz-smooth, which means the *gradient* is Lipschitz continuous. For instance, see 1) the third bullet point in their Section 1.1, b) the paragraph "Different loss functions" before Theorem 6, c) the first paragraph in their Appendix A. The precise condition needed is given in the footnote on the first page of the Appendix: "we assume that $f(z;y)$ is 1-Lipschitz (upper) smooth with respect to $z$" which means $f(z+z';y) \le f(z) + \langle \nabla f(z;y), z' \rangle + \frac{1}{2}|| z' ||^2$. It is easy to see that this follows from 1-Lipscthiz gradient assumption $||\nabla f(z+z') - \nabla f(z)|| \le ||z'||$. We think this is a central assumption in Allen-Zhu's proofs and cannot be omitted. Please correct us if we are wrong and/or if there are other references that we could check out that might be more suitable for our MSO setting.

---

> > > > ### Comment · Reviewer_4Grh · 2022-12-08
> > > > **agreed**
> > > >
> > > > Yes, you are right, that condition cannot be mis-interpreted (in spite of, to me, their odd choice of language). Sorry I was mistaken. This seems then to be an interesting open problem for future work...

---

### Official Review · Reviewer_HUdf · 2022-11-04

**Confidence:** 2
**Clarity, Quality, Novelty And Reproducibility:** The paper lacks clarity
**Correctness:** 3
**Technical Novelty And Significance:** 2
**Empirical Novelty And Significance:** 2
**Recommendation:** 5

**Strength And Weaknesses:**


Major concerns:
I do not know if it is because this paper is too far from my research area, but I really had troubles to understand the idea and the originality of the paper.

- I found the paper very verbose "the proximal term in our loss promotes convexity of the formulation". I am not sure to understand how the paper is related to Wasserstein graident flow. Is it only because they also parametrize le fixed-point oerator by a neural network?
- At the end of section 3.2 authors talk about computational gains, I did not understand were do these gains come from. Would it be possible to write the training algorithm, with the cost of each step in order to understand better? In the same vein, it cold be nice to make the difference with usual L20 method clearer, from myself I did not understand.

Experiments:
I cannot comment on all the experiments, but it feels like a succession of toy examples, without proving that the method is actually useful.
I can comment on the sparse recovery benchmark: why authors used an $\ell$-p norm, which is not alpha semi-convex, despite their exist alpha-semi-convex sparsity promoting norm such as the Minimax Concave Penalty, or SCAD, see for instance [1]. Morover this benchmark is performed in dimesion $d=8$, and $m=4$, which is too small to be considered as a serious experiment. For problem of this scale, it seems that one can directly solve the original $\ell_0$ constrained problem [2]

[1] Breheny, P. and Huang, J., 2011. Coordinate descent algorithms for nonconvex penalized regression, with applications to biological feature selection. The annals of applied statistics, 5(1), p.232.

[2] Marmin, A., Castella, M. and Pesquet, J.C., 2019, May. How to globally solve non-convex optimization problems involving an approximate ℓ 0 penalization. In ICASSP 2019-2019 IEEE International Conference on Acoustics, Speech and Signal Processing (ICASSP) (pp. 5601-5605). IEEE.



**Summary Of The Paper:**

Authors propose a new estimator, inspired from proximity operator theory.
The proposed estimator is supposed to be able to recover multiple local minima.
Authors proposed multiple experiments on toy problems


**Summary Of The Review:**

In conclusion I did not fully understand the idea and the contribution of the paper. The theoretical and practical impact of the paper seems limited.

---

> ### Author Response · Authors · 2022-11-08
> **Author's Response (1/2)**
>
> Thank you for your candid review. While we appreciate the thoughtful feedback, we remain confident that our work will be a strong contribution to the ICLR 2022 program. We are happy to share clarifications about our work during the discussion process and revise the text of our paper as needed.
>
> Please find below answers to questions raised in your initial review.
>
> **I do not know if it is because this paper is too far from my research area, but I really had troubles to understand the idea and the originality of the paper.**
>
> If possible, we would appreciate it if the reviewer can clarify which aspects of our work they found difficult to understand.
>
> We suggest focusing on the concrete example of object detection described in the third paragraph of the introduction, which demonstrates the originality and interest of our framework as described in section 3.
>
> **I found the paper very verbose "the proximal term in our loss promotes convexity of the formulation"**
>
> The “proximal term” refers to $\frac{\lambda}{2}||{\Phi(x,\tau) - x}||_2^2$ in (2), similar to the proximal term $\frac{\lambda}{2}||y-x||_2^2$ in (1). For (1), the “promotion of convexity” means the function $f_\tau(y) + \frac{\lambda}{2}||y-x||_2^2$ is more convex than $f_\tau(y)$: if you compute the Hessian you will find an additional $\frac{\lambda}{2} I$ term, which increases all eigenvalues of the resulting Hessian by $\frac{\lambda}{2}$. The same effect happens for (2) as well, as shown in the proof of Theorem 3.1.
>
> **I am not sure to understand how the paper is related to Wasserstein gradient flow.**
>
> ***We have clarified this in the new Section E in the revised appendix of the paper where we show that solving (2) is the same as solving the JKO discretization of Wasserstein gradient flows for linear functionals.***
>
> **At the end of section 3.2 authors talk about computational gains, I did not understand w[h]ere do these gains come from. Would it be possible to write the training algorithm, with the cost of each step in order to understand better? In the same vein, it co[u]ld be nice to make the difference with usual L20 method clearer**
>
> The computational gains that are referred to at the end of section 3.2 are for *test time*.
>
> Typical L2O methods require evaluating the objective $f_\tau$ or its gradient at test time (e.g. in equation (1) of [1], the gradient of the objective $f$ is needed), whereas in our method this is not needed. The objective $f_\tau$ can be expensive to evaluate (e.g. for symmetry detection in 5.4) or simply not available at test time (e.g. for object detection in 5.5).
>
> Many L2O methods can be thought of as learning parameters to improve an optimizer (e.g. searching for a better Adam) that in addition keep a summary of past gradients; these methods typically require sequence models such as LSTM to represent this history information. We gave a short description of the usual L2O methods in the first paragraph of Section D.3. To highlight our method from the existing L2O literature, **existing L2O methods are not designed to handle multiple solutions**. In Section D.3 we gave a construction of a counterexample to show why existing L2O methods fail to recover multiple solutions; we also verified this claim empirically (see Figure D.2). Moreover, to the best of our knowledge, none of the existing L2O methods has theoretical guarantees on the convergence of training.
>
> For training, we simply run stochastic gradient descent on (2) to optimize for the proximal operator $\Phi$. The details of $f_\tau$ for each application is given in Section 5.
>
> [1] Andrychowicz, Marcin, et al. "Learning to learn by gradient descent by gradient descent." Advances in neural information processing systems 29 (2016).

---

> ### Author Response · Authors · 2022-11-08
> **Author's Response (2/2)**
>
> **Experiments: I cannot comment on all the experiments, but it feels like a succession of toy examples, without proving that the method is actually useful**
>
> We strongly disagree with this comment.
>
> We include five applications, three of which (5.1, 5.2, 5.3) are smaller-scale examples, and two (5.4, 5.5) tackle large-scale practical applications (symmetry detection and object detection).
>
> We choose a diverse set of experiments to demonstrate the broad applicability of our method. The initial three applications employ  small-scale settings for which we can compute ground truth using particle descent reliably, needed to compute our witness metrics (Figure D.4, D.5, D.7, D.8). This does not mean the methodology proposed in 5.1-5.3 is restricted to “toy” examples. For instance, a straightforward extension of our method for sampling from level sets (5.1) can be used to design a 3D shape completion pipeline if we couple the objective in 5.1 with a point-cloud encoder like in 5.4.
>
> Moreover, our method shows promising results for the well-studied problem of object detection. Our numbers are comparable with highly optimized Faster-RCNN (with 40% less network parameters), *despite not having any highly-specialized architectural design*. Crucially, *we do not need to predict a fixed number of bounding boxes*, which is novel and advantageous as compared to the baseline FN in Table 1. We speculate that with more specialized architectures, our method can advance state-of-the-art object detection, although detailed experiments in this direction are out of the scope of our current discussion.
>
> **I can comment on the sparse recovery benchmark: why authors used an $\ell$-p norm, which is not alpha semi-convex, despite their exist alpha-semi-convex sparsity promoting norm such as the Minimax Concave Penalty, or SCAD**
>
> We chose $\ell_p$ norms since we design it to be a challenging case as varying $p$ drastically changes the convexity of the optimization landscape. For $\ell_p$ norms, our method obtains strong performance even against strong baseline such as proximal gradient descent with closed-form thresholding formula (Figure D.8).
>
> ***We have added a new experiment at the end of Section D.5 on sparse recovery with minimax concave penalty (MCP) as you suggested***. In Figure D.9, our method obtains strong performance consistent with that of $\ell_p$ norms. We emphasize that the goal of the paper is to propose a general framework for solving MSO, and *sparse recovery is only one application of the much more general formulation we propose*.

---

> ### Author Response · Authors · 2022-11-14
> **Could you please check our response?**
>
> Dear Reviewer,
>
> Since only a few days remain in the discussion period, we would appreciate it if you check and reply to our response to your comments soon. This will give us time to address further questions and comments that you may have before the end of the discussion period. If our response adequately addresses your concerns, please consider raising the score of our submission. Thank you very much for your time.

---

> ### Author Response · Authors · 2022-12-08
> **Could you please check our response?**
>
> Dear Reviewer,
>
> As we are approaching the end of the discussion period, we would appreciate it if you check and reply in this thread, especially since it seems like there is some misunderstanding that we have hopefully clarified in our response and in the revised paper. If our response adequately addresses your concerns, please consider raising the score of our submission. Thank you very much for your time.

---

### Decision · Program_Chairs · 2023-01-20

**Decision:**

Accept: poster

**Justification For Why Not Higher Score:**

The paper has some clarity issues. The approach doesn't work in high dimensions.

**Justification For Why Not Lower Score:**

The paper will be of interest to the learning-to-optimize community.

**Metareview: Summary, Strengths And Weaknesses:**

This paper proposes a method to learn proximal mappings which have multiple beneficial properties as optimisers.

Strengths:
- The authors present a method applicable to general (multi-solution optimization) MSO problems
- Theoretical guarantees in learning to optimize are rare
- A variety of experiments illustrating the method

Weaknesses:
- The paper is sometimes hard to follow, clarity could be improved
- The method may not work well in higher dimension

2 reviewers out of 3 agree that the paper is a strong accept. 1 reviewer was more negative, mainly pointing clarity issues.

I recommend acceptance as a poster.

**Note From Pc:**

if the above contains the word "oral" or "spotlight" please see: "oral" presentation means -> notable-top-5% and "spotlight" means -> notable-top-25%. As stated in our emails, we are disassociating presentation type from AC recommendations